# HONet: Data-Efficient Learning for Exact Cover Tasks via Hypergraph Optimization

**Pengyang Huang** [1]  **Zirui Zhuang** [1]  **Haifeng Sun** [1]  **Qi Qi** [1]  **Jingyu Wang** [2 1]  **Jianxin Liao** [1]

## Abstract

Deep learning approaches typically require prohibitive amounts of data to approximate known-constraint Exact Cover tasks, while existing neuro-symbolic methods often face training infeasibility and scalability bottlenecks. To bridge this divide, we propose the Hypergraph Optimization Network (HONet), an end-to-end framework integrating a structure-preserving Deep Residual Hypergraph Encoder with a differentiable fixed-constraint Quadratic Programming layer. By adopting a "Fixed Polytope" paradigm guided by the Geometric Consistency Loss, HONet explicitly shapes the objective landscape, encouraging the valid discrete solution to align with the global energy minimum. Empirical results show that HONet rapidly achieves 100% accuracy on $9 \times 9$ Sudoku using limited samples, exhibiting superior data efficiency over baselines while maintaining exceptional robustness in highly sparse regimes and additional tasks.

## 1. Introduction

Predicting discrete structures characterized by intricate high-order dependencies is a core challenge in Structured Output Prediction (Belanger & McCallum, 2016); Exact Cover Problems (ECPs) form an important class of such settings (Karp, 2009), requiring the exact satisfaction of covering constraints, where a solution selects local candidates so that each constraint item is covered exactly once. While traditional symbolic solvers guarantee correctness once the constraints are specified (De Moura & Bjørner, 2008; Davis & Ji, 2025), their non-differentiable nature makes end-to-end

optimization with raw sensory inputs difficult. Conversely, Deep Learning excels in intuitive perception but can struggle to satisfy the strict, discrete logical rules inherent in exact structural assignment (Colelough & Regli, 2025).

Recent research bridges this divide via several primary paradigms. The first, data-driven approaches, ranges from domain-specific recurrent architectures (Yang et al., 2023) to Large Language Models (LLMs) employing Chain-of-Thought (CoT) (Hao et al., 2023). These methods typically learn rule satisfaction implicitly from examples rather than enforcing the known constraint topology as an explicit feasible set. Consequently, data requirements can be substantial in hard tasks; for instance, Recurrent Relational Networks (Palm et al., 2018) require over 100k labeled examples to master Sudoku, while LLMs may suffer from logical hallucinations due to their probabilistic autoregressive nature (Dziri et al., 2023). Without explicit feasibility enforcement, such models can still yield infeasible solutions that require post-hoc correction or search (Giannoulis et al., 2025).

The second paradigm, optimization-based neuro-symbolic learning, addresses validity by embedding structured reasoning layers into neural architectures. General-purpose differentiable solvers such as OptNet (Amos & Kolter, 2017) and CvxpyLayers (Agrawal et al., 2019) enable gradients to flow through quadratic or conic programs, allowing discrete structured tasks to be trained via continuous relaxations (Wilder et al., 2019). However, such adaptations may produce fractional solutions, requiring additional mechanisms to recover discrete assignments. Furthermore, these methods may face numerical instability and primal infeasibility during learning when the constraint structure is learned or neural-parameterized (Bambade et al., 2024). Specific adaptations like SATNet (Wang et al., 2019) mitigate scalability via SDP-based relaxations, but vectorized representations may obscure the native high-order topological structure (Chang et al., 2020). Related decision focused learning methods and surrogate losses train predictors through downstream optimization objectives (Mandi et al., 2024; Zhao et al., 2025; Elmachtoub & Grigas, 2022), but they primarily target decision quality under uncertain objectives rather than exact completion under known symbolic constraints.

Against this backdrop, we propose the Hypergraph Opti-

---

[1]State Key Laboratory of Networking and Switching Technology, Beijing University of Posts and Telecommunications, Beijing 100876, China [2]Guilin University of Electronic Science and Technology, Guangxi 541004, China. Correspondence to: Zirui Zhuang <zhuangzirui@bupt.edu.cn>, Jingyu Wang <wangjingyu@guet.edu.cn>.

*Proceedings of the $43^{rd}$ International Conference on Machine Learning*, Seoul, South Korea. PMLR 306, 2026. Copyright 2026 by the author(s).

mization Network (HONet), an end-to-end differentiable framework for known-constraint exact-cover tasks. In these tasks, a shared constraint template provides the global reasoning structure, while instance-specific clues or observations restrict the candidate completions within that structure. Challenging the inefficient "tabula rasa" learning paradigm for tasks with these known structural priors, HONet redefines the division of labor: rather than forcing neural networks to rediscover explicit constraints, it separates neural perception from symbolic feasibility and lets neural networks focus on their strength—intuitive perception:

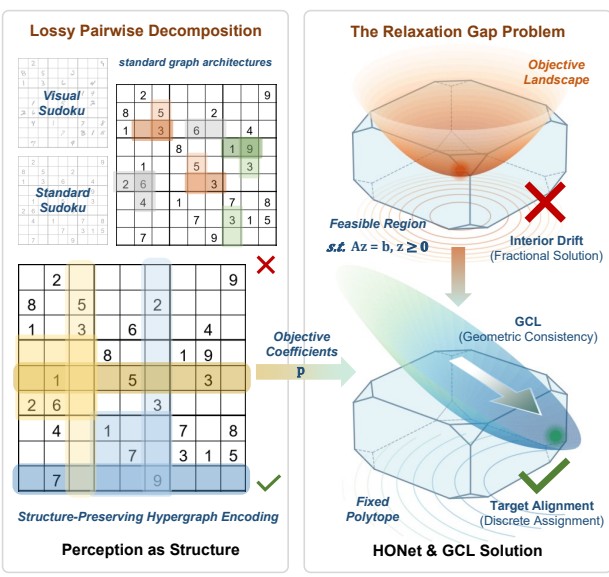

*Figure 1.* Overcoming structural and geometric barriers. **Left:** Standard pairwise graph architectures decompose dependencies into binary relations via clique expansion, which may discard high-order scope information. Our hypergraph encoder preserves the variable–constraint incidence structure. **Right:** Relaxing discrete constraints into a continuous feasible polytope can introduce a Relaxation Gap. HONet, guided by GCL, learns objective coefficients **p** to reshape the objective landscape, encouraging the relaxed optimum to align with the target discrete assignment.

**Perception as Structural Optimization Modeling.** To address the information loss of standard pairwise graph architectures (Kipf, 2016) in representing high-order constraint structure (Figure 1), we employ a Deep Residual Hypergraph Neural Network (HGNN) (Feng et al., 2019). This choice is motivated by its ability to preserve the native variable–constraint incidence structure of exact-cover formulations (Kschischang et al., 2002), as well as by residual connections that support long-range constraint propagation for global consistency. Unlike "rule discovery" paradigms that may suffer from infeasibility or instability when learning constraints from scratch, the HGNN provides a structural inductive bias for "rule parameterization": transforming raw

inputs directly into the linear coefficients of a latent objective function. This avoids reallocating model capacity to relearn known structural rules and lets the fixed decoder handle feasibility over the relaxed exact-cover region.

**Reasoning via Geometric Alignment.** Specifically, we decode these known-constraint exact-cover tasks with a fixed convex QP. To mitigate the Relaxation Gap, in which solutions may drift toward the interior of the feasible polytope (Figure 1), we view the neural network as a "objective landscape shaper." Leveraging the KKT optimality conditions (Boyd & Vandenberghe, 2004) as the theoretical bridge between learned objective coefficients and the target assignment, we introduce the Geometric Consistency Loss (GCL). By combining coordinate-level alignment with a regularized energy-gap term, GCL encourages consistency in the underlying optimization geometry, guiding the network to reshape the landscape and mitigate the Relaxation Gap.

We evaluate HONet on the $9 \times 9$ Sudoku benchmark as a representative known-constraint exact-cover completion task. The results demonstrate that HONet achieves substantial data-efficiency gains: it reaches 100% accuracy with only 9k training samples and remains robust when the training set is reduced to 1%. Its performance is competitive with baselines trained on substantially more data, while achieving zero-shot OOD generalization on extremely sparse test sets (17-19 Givens). We further quantitatively validate the effectiveness of GCL by analyzing the dynamics of the Relaxation Gap and relaxed-solution metrics. In the Visual Sudoku task, HONet extends the same fixed-feasibility framework to perceptual inputs, converging faster than SAT-Net (Wang et al., 2019) in our experiments. Finally, we probe the boundaries of the framework on the Futoshiki task, revealing the geometric mismatch that arises when task-defining inequalities fall outside the fixed solver formulation. In summary, our main contributions are as follows:

- **A Novel Neuro-symbolic Framework for Known-constraint Exact Cover.** We propose HONet, an end-to-end architecture integrating a structure-preserving Deep Residual HGNN with a differentiable fixed-constraint QP layer. It shifts the paradigm from "implicit rule discovery" to "explicit rule parameterization," ensuring relaxed feasibility and strong data efficiency for known-constraint exact cover tasks.

- **Geometric Consistency Loss.** Addressing the Relaxation Gap, we introduce GCL. Theoretically motivated by the KKT optimality conditions, this loss combines coordinate-level alignment with a regularized energy-gap term, guiding the neural network to reshape the objective landscape and mitigate the interior drift issue.

- **Theoretical & Empirical Validation.** We provide a mechanism analysis through the lens of optimization

geometry, clarifying how the fixed QP and GCL relate learned objective coefficients to relaxed solutions. Extensive experiments analyze the observed effectiveness and data efficiency associated with these relations, and reveal the boundary of the fixed-polytope framework when task-defining constraints are instance-varying.

## 2. Related Works

**Hypergraph-based Combinatorial Optimization.** Hypergraphs provide a natural representation for high-order constraints in neural optimization. While traditional parallel algorithms (Dhulipala et al., 2024) offer approximation guarantees for symbolic Set Cover, they do not address end-to-end perception. Conversely, neural solvers like Deep $k$-grouping (Bai et al., 2025) utilize hypergraph biases via unsupervised OH-PUBO, yet may remain susceptible to local optima under penalty-based relaxations. HONet uses the hypergraph not as a standalone neural solver, but as a structure-preserving encoder for latent objective coefficients, and couples it with a downstream QP decoder.

**Differentiable Optimization Layers.** This line of work facilitates structured reasoning by embedding mathematical programming directly into neural architectures. OptNet (Amos & Kolter, 2017) and CvxpyLayers (Agrawal et al., 2019) established the foundations for differentiating through quadratic and general convex programs, respectively. This paradigm was further extended to logic-based reasoning via SATNet (Wang et al., 2019), which employs SDP relaxation for MAXSAT, and SMTLayer (Wang et al., 2023), which grounds neural inference within Satisfiability Modulo Theories. To address numerical instability, Bambade et al. (2024) leveraged augmented Lagrangian techniques to enable gradient backpropagation even through infeasible QP layers. In contrast to approaches that learn or neural-parameterize constraints, HONet instead studies objective-landscape learning in a complementary known-constraint regime.

**Decision-Focused Learning and Projection Layers.** Our work is closely related to DFL, which trains predictive models through downstream optimization (Mandi et al., 2024). Paradigms such as SPO+ (Elmachtoub & Grigas, 2022) focus on minimizing decision regret under fixed constraints, often learning semantically meaningful optimization parameters (Kotary et al., 2023; Shah et al., 2022; Mandi et al., 2022). HONet shares a similar structure but focuses on learning latent objective coefficients guided by GCL. The QP decoder is also related to projection-based constrained learning. However, projection layers are often used as safety or repair mechanisms (Chen et al., 2021), minimally correcting neural decisions to satisfy constraints. In HONet, the fixed polytope encodes the task's symbolic feasibility. The neural module therefore shapes an objective landscape over this fixed region to guide solution selection.

## 3. Methodology

### 3.1. Problem Formulation

We consider finite-domain assignment problems with known structural constraints. Let $\mathcal{V}$ denote $n$ variables, and let $\mathcal{D} = \{l_1, \ldots, l_d\}$ represent a finite set of discrete labels. We represent variable assignments as an indicator matrix $\mathbf{Z} \in \{0,1\}^{n \times d}$ and its flattened vector $\mathbf{z} = \mathrm{vec}(\mathbf{Z}) \in \{0,1\}^N$, where $N = n \times d$. A valid assignment must satisfy the simplex constraint $\sum_{j=1}^d \mathbf{Z}_{i,j} = 1$ for each variable $i$, alongside hard structural constraints. Focusing on exact-cover-style rules encodable as linear equalities, we define the discrete feasible domain $\Omega$ as:

$$\Omega = \left\{ \mathbf{z} \in \{0,1\}^N \mid \mathbf{A}\mathbf{z} = \mathbf{b}, \sum_{j=1}^d \mathbf{Z}_{i,j} = 1, \forall i \right\}, \quad (1)$$

where $\mathbf{A}$ and $\mathbf{b}$ encode the known structural rules. The simplex constraints can also be absorbed into $\mathbf{A}\mathbf{z} = \mathbf{b}$; we write them separately for clarity. As direct optimization over the discrete feasible set is non-differentiable, we reformulate solution selection as a parameterized optimization problem. Given objective coefficients $\mathbf{p} \in \mathbb{R}^N$ output by the neural network, we interpret the discrete ground truth $\mathbf{z}^*$ not merely as a feasible solution, but as a target assignment to be favored within the objective landscape shaped by $\mathbf{p}$:

$$\mathbf{z}^* \in \arg\min_{\mathbf{z} \in \Omega} E_{\mathrm{lin}}(\mathbf{z}; \mathbf{p}), \qquad E_{\mathrm{lin}}(\mathbf{z}; \mathbf{p}) = -\mathbf{p}^\top \mathbf{z}. \quad (2)$$

This reformulation transforms discrete solution selection into learning objective coefficients over a known feasible region, setting up the relaxed differentiable optimization layer introduced below.

### 3.2. Structural Optimization Modeling

As defined in Section 3.1, our core task is to shape the objective coefficients $\mathbf{p}$ over a known exact-cover feasible region, thereby constructing the objective landscape. To achieve this, we require an encoder capable of capturing the high-order dependencies of Exact Cover problems, with a structure-preserving inductive bias over the native variable–constraint incidence structure.

Topologically, each structural constraint scope $S_k \subseteq \mathcal{V}$ acts on a local subset of variables and corresponds to a high-order relation among them. At the exact-cover equality level, such scopes induce local exact-one functions over the corresponding assignment coordinates. To capture this, HONet maps variables to nodes $\mathcal{V}$ and constraint scopes to hyperedges $\mathcal{E}_H$, explicitly connecting the variables within

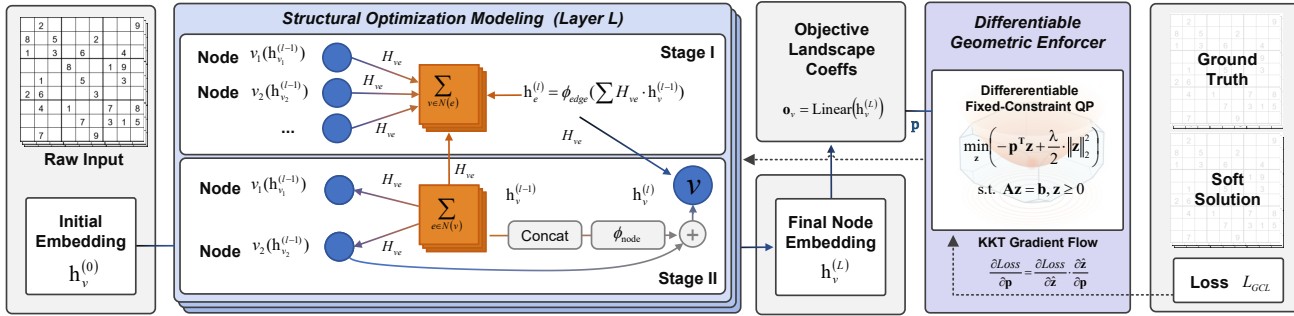

*Figure 2.* **Overview of the HONet Architecture.** Deep Residual Encoder: Raw inputs are projected to $\mathbf{h}_v^{(0)}$. The encoder alternates between Constraint Aggregation (utilizing normalized incidence entries $H_{ve}$ and 2-layer MLPs with LayerNorm) and Variable Update (via $\phi_{\text{node}}$ and residual connections). Final embeddings are scaled by temperature $\tau$ to form objective coefficients $\mathbf{p}$, which regulate the steepness of the objective landscape. This landscape guides the QP layer to compute the relaxed optimum, trained end-to-end via GCL.

each scope $S_k$. This representation preserves the native identity of high-order constraint scopes. By contrast, let $G_{\text{clq}}(\mathcal{V}, \mathcal{E}_H)$ denote the untyped clique-expanded pairwise encoding of $(\mathcal{V}, \mathcal{E}_H)$, where two variables are connected iff they co-occur in some hyperedge $e \in \mathcal{E}_H$. Such an encoding keeps only pairwise co-occurrence relations and may discard the original hyperedge identity/incidence information. This structural distinction is formalized as follows:

**Proposition 3.1.** *Consider an abstract binary exact-cover hypergraph $(\mathcal{U}, \mathcal{E})$, where $\mathcal{U}$ indexes binary assignment coordinates. For any exact-cover constraint $c_k$ with scope $S_k \subseteq \mathcal{U}$ and associated local function $\Psi_k$, $\Psi_k$ is permutation-invariant on $S_k$. In particular, there exists a permutation-invariant hyperedge aggregation form that preserves the native scope $S_k$ and can represent $\Psi_k$ on the binary assignment domain. On the other hand, the untyped clique-expanded pairwise encoding map $(\mathcal{U}, \mathcal{E}) \mapsto G_{\text{clq}}(\mathcal{U}, \mathcal{E})$ is non-injective with respect to native hyperedge identity. (Proof provided in Appendix.)*

**Deep Residual Encoding.** To enable the long-range constraint propagation needed for global consistency and to support iterative structured reasoning, we construct a deep encoder based on spatial message passing (Gilmer et al., 2017). Let $\mathbf{h}_v^{(l)}$ and $\mathbf{h}_e^{(l)}$ denote the states of variable node $v \in \mathcal{V}$ and hyperedge $e \in \mathcal{E}_H$ at layer $l$, respectively. Each layer performs a two-stage variable–constraint update inspired by factor-graph message passing (Yedidia et al., 2003):

**Stage I: Constraint aggregation (Variable-to-Factor).** Hyperedges aggregate messages from their incident variable nodes, functionally analogous to the variable-to-factor direction in factor-graph message passing:

$$\mathbf{h}_e^{(l)} = \phi_{\text{edge}} \left( \sum_{v \in \mathcal{N}(e)} H_{ve} \cdot \mathbf{h}_v^{(l-1)} \right), \qquad (3)$$

where $H_{ve}$ denotes the hypergraph incidence weight and $\phi_{\text{edge}}$ is an MLP with layer normalization that produces the hyperedge representation.

**Stage II: Variable update (Factor-to-Variable).** Variables update their states by aggregating feedback from incident hyperedges. We employ residual connections to support stable deep message passing and gradual feature refinement:

$$\mathbf{h}_v^{(l)} = \mathbf{h}_v^{(l-1)} + \phi_{\text{node}} \Big( \text{Concat} \big[ \mathbf{h}_v^{(l-1)}, \sum_{e \in \mathcal{N}(v)} H_{ve} \mathbf{h}_e^{(l)} \big] \Big). \qquad (4)$$

This architecture injects a structural inductive bias by routing information along the variable–constraint incidence structure. After $L$ layers, each variable-node embedding is mapped to label-wise potentials:

$$\mathbf{o}_v = \text{Linear}(\mathbf{h}_v^{(L)}) \in \mathbb{R}^d, \quad v \in \mathcal{V}. \qquad (5)$$

The QP objective coefficients are then obtained by temperature scaling and flattening:

$$\mathbf{p} = \frac{1}{\tau} \cdot \text{vec} \left( \{\mathbf{o}_v\}_{v \in \mathcal{V}} \right) \in \mathbb{R}^N. \qquad (6)$$

In practice, $\tau$ is annealed during training to gradually sharpen the learned objective landscape. This completes the parameterized construction of the objective function and yields the instance-specific objective coefficients used by the differentiable QP layer introduced next.

### 3.3. Differentiable Geometric Enforcer

Directly optimizing over the discrete feasible set $\Omega$ is non-differentiable, while standard linear relaxations can have non-smooth solution mappings that are difficult to use for stable end-to-end learning. To enable gradient-based training, we adopt the paradigm of differentiable optimization

(Amos & Kolter, 2017). We relax the binary domain to a continuous polytope $\mathcal{P} = \{\mathbf{z} \in \mathbb{R}^N \mid \mathbf{A}\mathbf{z} = \mathbf{b}, \mathbf{z} \geq 0\}$, and formulate inference as a strongly convex QP:

$$\hat{\mathbf{z}} = \arg\min_{\mathbf{z} \in \mathcal{P}} \left( -\mathbf{p}^\top \mathbf{z} + \frac{\lambda}{2}\|\mathbf{z}\|_2^2 \right). \qquad (7)$$

The quadratic regularization term imparts curvature, making the relaxed objective strongly convex and yielding a unique solution for each $\mathbf{p}$ when $\mathcal{P}$ is nonempty. Implemented via CvxpyLayers (Agrawal et al., 2019), this layer computes the solution sensitivity $\frac{\partial \hat{\mathbf{z}}}{\partial \mathbf{p}}$ by differentiating through the KKT conditions, enabling gradients to the upstream encoder.

**"Fixed Polytope, Dynamic Objective" Paradigm.** Crucially, we treat $\mathbf{A}$ and $\mathbf{b}$ as immutable constants derived from the problem definition, rather than learnable parameters. This decouples symbolic feasibility from feature learning, ensuring the polytope $\mathcal{P}$ remains fixed throughout training while instance-specific information enters through the learned objective coefficients $\mathbf{p}$. Functionally, this layer acts as a geometric enforcer, guiding the learned objective landscape to be optimized over the fixed relaxed feasible region defined by the known exact-cover rules.

### 3.4. Geometric Consistency Learning

**Geometric Gap and Normal-Cone Regions.** Before detailing the loss, we analyze the relationship between the learned objective coefficients $\mathbf{p}$ and the solver's output $\hat{\mathbf{z}}$. We define the regularized energy used by the QP layer as $\mathcal{E}_\lambda(\mathbf{z}; \mathbf{p}) = -\mathbf{p}^\top \mathbf{z} + \frac{\lambda}{2}\|\mathbf{z}\|_2^2$, and define the Relaxation Gap as the regularized energy difference between the discrete ground truth $\mathbf{z}^*$ and the relaxed solution $\hat{\mathbf{z}}$. From the KKT optimality conditions, $\mathbf{z}^*$ is optimal over the fixed polytope $\mathcal{P}$ whenever the learned coefficients satisfy the normal-cone condition $\mathbf{p} - \lambda\mathbf{z}^* \in N_\mathcal{P}(\mathbf{z}^*)$. This shows that the desired coefficients need not be a precise point estimate; instead, there is a geometric region of coefficients that can make the target assignment favorable under the fixed QP. However, to encourage the solver to move toward the target assignment, the learned coefficients $\mathbf{p}$ should have sufficient magnitude and proper direction to counteract the regularization shrinkage. This motivates a loss function that explicitly minimizes this regularized Relaxation Gap, rather than merely supervising label accuracy (see Appendix for full derivation).

**Geometric Consistency Loss.** To fully leverage this mechanism, we propose GCL, combining coordinate-level alignment with regularized energy-gap guidance:

$$\mathcal{L}_{GCL} = \alpha \cdot \mathcal{L}_{MSE} + \beta \cdot \mathcal{L}_{gap} + \gamma \cdot \mathcal{L}_{warm}. \qquad (8)$$

**Coordinate-Level Anchor ($\mathcal{L}_{MSE}$).** We include $\mathcal{L}_{MSE} = \|\mathbf{z}^* - \hat{\mathbf{z}}\|_2^2$ to provide direct gradient flow toward the coordinate truth, encouraging coordinate-level consistency between the relaxed solution and the target assignment.

**Relaxation Gap Loss ($\mathcal{L}_{gap}$).** This term provides an energy-level geometric signal, encouraging the discrete target assignment $\mathbf{z}^*$ to have lower regularized energy than the relaxed solution $\hat{\mathbf{z}}$ in the objective landscape:

$$\mathcal{L}_{gap} = \text{ReLU}\left(\mathcal{E}_\lambda(\mathbf{z}^*; \mathbf{p}) - \mathcal{E}_\lambda(\hat{\mathbf{z}}; \mathbf{p})\right). \qquad (9)$$

Minimizing $\mathcal{L}_{gap}$ explicitly reshapes $\mathbf{p}$ so that $\mathbf{z}^*$ becomes energetically favorable within the relaxed polytope.

**Auxiliary Warm-Start Supervision ($\mathcal{L}_{warm}$).** While the coordinate-level and energy-level losses are effective for standard tasks, convergence can become more challenging under sparse-clue settings. In such cases, the instance-specific evidence for selecting the target completion can be weak, making it harder for QP-level losses alone to shape a favorable objective landscape. Therefore, we introduce $\mathcal{L}_{warm} = \frac{1}{|\mathcal{V}|}\sum_{v \in \mathcal{V}} \text{CE}(\mathbf{o}_v, y_v^*)$ as an auxiliary warm-start term acting directly on the label-wise potentials before QP decoding. It provides direct task-level supervision for the neural potentials and complements the coordinate-level and energy-level losses. See Appendix for details on its implementation and role in sparse-clue settings.

## 4. Experiments

We evaluate HONet on the $9 \times 9$ Sudoku benchmark, a representative exact-cover task, to address three core questions: **RQ1 (Data Efficiency):** Can HONet leverage known constraints to achieve strong performance with markedly fewer samples than neural baselines? **RQ2 (OOD Robustness):** Does the model generalize to unseen sparsity configurations? **RQ3 (Mechanism Verification):** Do the Hypergraph Encoder and GCL effectively mitigate the Relaxation Gap?

Furthermore, we extend evaluations to Visual Sudoku and Futoshiki to probe the framework's adaptability to perceptual input and the boundaries of handling hybrid constraints.

### 4.1. Experimental Setup

**Datasets.** We evaluate performance on the $9 \times 9$ Sudoku benchmark, stratified into five difficulty levels according to the number of Givens. For each level, we construct datasets containing 9k training, 1k validation, and 1k test samples, drawn from SATNet ($\sim 36.2$ Givens) (Wang et al., 2019), Kaggle (Medium/Hard, 20–28 Givens) (Park, 2018), and RRN (Extreme, 17–19 Givens, near the minimal-clue regime) (Palm et al., 2018). For the $\sim 36.2$ Givens setting, we construct the validation split using the OptNet Sudoku generation script (Amos & Kolter, 2017).

**Baselines.** We compare against three classes of solvers:

*Table 1.* Test results on the $9 \times 9$ Sudoku benchmark. We report Whole-board Accuracy (%) across difficulty levels. "Time" denotes the average inference time per puzzle on the test set, and "Epochs" denotes the epoch at which peak validation performance was reached. "CNN" refers to the model without iterative inference at test time. HONet uses an inference temperature of 0.1.

| DATA SIZE | | **HONET** **9K** | BACKTRACK - | Z3 SOLVER - | RRN 9K | 180K | CNN+ITER 1M | CNN 1M | SATNET 9K |
|---|---|---|---|---|---|---|---|---|---|
| S. 36.2G. | ACC. | **100.00%** | 100.00% | 100.00% | 99.50% | 100.00% | 100.00% | 18.40% | 98.60% |
| | TIME | 10.94MS | 2.09MS | 33.70MS | 1.05MS | 1.26MS | 17.20MS | **0.25MS** | 5.36MS |
| | EPOCHS | **1** | - | - | 22 | 1 | 57 | 66 | 99 |
| 26-28 G. | ACC. | **100.00%** | 100.00% | 100.00% | 56.90% | 99.10% | 61.10% | 0.00% | 0.00% |
| | TIME | 11.45MS | 532.33MS | 163.62MS | **2.68MS** | 3.58MS | 15.73MS | - | - |
| | EPOCHS | **4** | - | - | 70 | 85 | 102 | - | - |
| 23-25 G. | ACC. | **100.00%** | 100.00% | 100.00% | 48.80% | 99.10% | 51.60% | 0.00% | 0.00% |
| | TIME | 54.67MS | 2.20S | 263.46MS | **2.70MS** | 3.67MS | 16.43MS | - | - |
| | EPOCHS | **5** | - | - | 47 | 87 | 115 | - | - |
| 20-22 G. | ACC. | **100.00%** | 100.00% | 100.00% | 33.90% | 98.60% | 41.80% | 0.00% | 0.00% |
| | TIME | 48.45MS | 11.85S | 419.56MS | **2.67MS** | 3.68MS | 17.42MS | - | - |
| | EPOCHS | **7** | - | - | 52 | 93 | 148 | - | - |

(1) Symbolic Solvers: Backtracking and Z3 (non-learned reference solvers) (De Moura & Bjørner, 2008); (2) Neuro-symbolic: SATNet, an optimization-based counterpart; (3) Pure Neural: RRN (a strong baseline for message passing methods) and an engineered Iterative CNN (Park, 2018).

To assess data efficiency, we establish two regimes: **Low-Data** (9k samples, same as HONet) and **High-Data** (baselines using 180k–1M samples). We exclude LLMs due to non-differentiable inference and focus instead on classic specialized architectures.

**Metrics & Implementation.** We report Whole-board Accuracy and Relaxation Gap (defined in Section 3.4). We also include Global Fuzziness, measuring the fractionality of the relaxed solution before discretization: $f = \frac{1}{N} \sum_{i=1}^{N} 4\hat{z}_i(1 - \hat{z}_i)$, where 0 indicates a perfect 0/1 integer solution. HONet is implemented in PyTorch with CvxpyLayers (Agrawal et al., 2019); the experimental setup is detailed in the Appendix.

### 4.2. Evaluation of Data Efficiency and Effectiveness

This subsection aims to answer RQ1 through quantitative comparisons across different data regimes. We first evaluate performance under the standard setting, followed by probing the lower bound of data efficiency via extreme scarcity tests.

Table 1 reveals a clear data-efficiency gap. Under the Low-Data Regime (9k), baseline models expose notable bottle-necks: SATNet and CNN without iterative inference drop to 0.00% accuracy on Medium/Hard datasets, while RRN ranges between 33.9%–56.9%. This suggests that 9k samples are insufficient for these baselines to statistically fit high-entropy constraints in harder Sudoku settings. In contrast, HONet maintains 100.00% accuracy on Medium/Hard datasets. Notably, HONet (9k) even outperforms High-Data

baselines on the reported splits, surpassing RRN trained on 180k samples and CNN-based baselines trained on 1M samples, while converging within just 1–7 epochs.

This efficiency stems from a paradigm shift: baselines rely on implicit rule discovery from statistics, whereas HONet leverages a fixed geometric prior for efficient rule parameterization. By decoupling structural validity from feature learning, HONet reduces the effective hypothesis space from a complex "Logic-Perception Joint Space" to a focused "Objective Parameter Space," enabling the model to focus on shaping the objective landscape even with limited samples.

While CPU-based convex optimization incurs higher latency than pure neural methods, HONet relies on polynomial-time convex optimization, in contrast to the exponential worst-case of backtracking search. Benefiting from the loss function's optimization of the Relaxation Gap, the relaxed solutions become sharper and more aligned with the target assignment; this correlation will be examined later.

Figure 3 probes the low-data limit of HONet's data efficiency. On Medium/Hard datasets, HONet converges rapidly, maintaining $> 99\%$ accuracy with only 500 samples ($\sim 5\%$), and reaching $\sim 90\%$ with just 100 samples ($\sim 1\%$). These results suggest that the geometric enforcer substantially reduces data dependence, enabling accurate structural assignment even under extreme low-data settings.

### 4.3. Robustness and Extreme Sparsity

This subsection aims to answer RQ2. We design two controlled experiments to evaluate the model's generalization robustness in Out-of-Distribution (OOD) scenarios and its performance boundaries when approaching the minimal-clue regime of Sudoku (17–19 Givens).

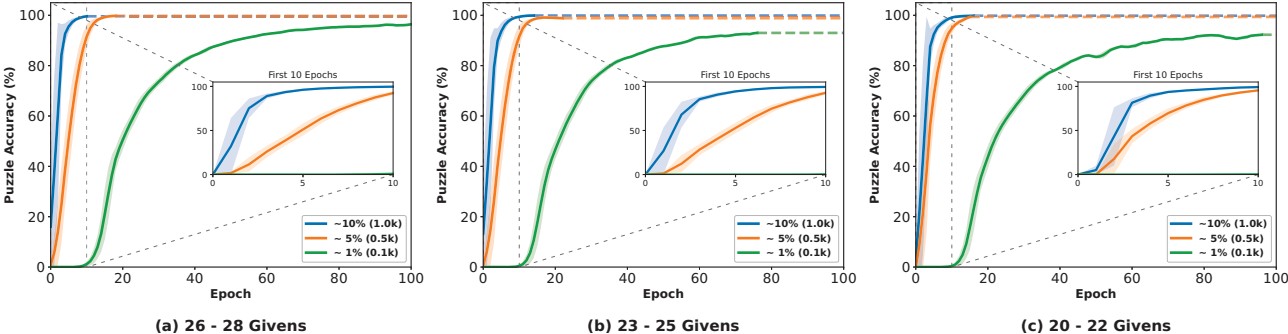

*Figure 3.* Data Efficiency Analysis. Accuracy curves on Medium/Hard datasets under extreme scarcity settings. We reduce the training set to $\sim 10\%$ (1k), $\sim 5\%$ (500), and $\sim 1\%$ (100) of the standard setting. The curves illustrate robust convergence even when data volume is reduced by up to nearly two orders of magnitude.

*Table 2.* Performance in the near-minimal-clue regime (17–19 Givens). We compare HONet with RRN and Recurrent Transformer ([Yang et al., 2023](#)) on 17–19 Givens tasks. HONet substantially outperforms 9k neural baselines and approaches baselines trained with 180k samples. For this sparse setting, HONet uses the corresponding inference $\lambda$ configuration; for other difficulty levels, we keep a conservative $\lambda = 1$ to maintain stable QP differentiation.

| | **OURS** | **RRN** | | **R. TRANS.** | |
|---|---|---|---|---|---|
| SIZE | **9K** | 180K | 9K | 180K | 9K |
| ACC. | 91.80% | 95.60% | 0.00% | 96.70% | 78.20% |

**OOD Generalization.** To evaluate robustness, we adopt a challenging "Single-Distribution Training, All-Distribution Testing" protocol. The model is trained exclusively on the "20–22 Givens" dataset (9k) and then evaluated in a zero-shot manner directly on test sets of all other difficulty levels. Results (see Appendix) show that the model exhibits strong bidirectional generalization: it maintains 100.00% accuracy on all easier/same-distribution datasets and achieves 87.50% on the unseen extreme dataset (17–19 Givens). This suggests that HONet learns transferable objective-shaping behavior under the shared Sudoku constraints. We observe that inference time increases in OOD scenarios, which may reflect greater difficulty in aligning the relaxed solution with the target assignment under distribution shifts.

**Near-Minimal-Clue Regime.** We focus on the extreme sparse setting of 17–19 Givens, where instance-specific evidence is limited and solution selection becomes more challenging (Table 2). In this regime, using the standard setting $\lambda = 1$ can make the quadratic regularization overly dominant. When bypassing the neural network and directly using the observed clues as $\mathbf{p}$ values for the convex optimization layer, forward accuracy is only $\sim 3\%$, indicating that sparse clues alone are insufficient to shape a favorable objective landscape. Without $\mathcal{L}_{warm}$, training with only the stan-

dard $\mathcal{L}_{MSE}$ and $\mathcal{L}_{gap}$ leads to a brief initial loss decrease followed by an early plateau at a suboptimal solution. To address this, we reduce $\lambda$ to attenuate the smoothing term and use $\mathcal{L}_{warm}$ to provide direct supervision on the label-wise potentials under sparse observations (see Section 3.4).

Under this optimized configuration, HONet achieves 91.80% accuracy using only 9k samples. In contrast, RRN (0.00%) and Recurrent Transformer (78.20%) suffer substantial degradation under the same data regime. HONet's performance with limited data approaches baselines trained on $20\times$ more data (180k), highlighting the effectiveness of its inductive bias.

### 4.4. Mechanism Analysis and Ablation Study

We dissect the contributions of HONet's core components via fine-grained ablation studies (Table 3) and delineate the neuro-symbolic boundary using non-learning baselines.

We compare HONet with Graph Transformer (GT)([Yun et al., 2019](#)) and Global Transformer ([Vaswani et al., 2017](#)) variants. As detailed in Table 3, HONet exhibits superior convergence efficiency, reaching peak performance in just 4 epochs under "26–28 Givens," whereas GT requires 18 epochs (see Appendix for convergence dynamics). This suggests that the 1-hop hyperedge pathway maps local constraint interactions directly to objective coefficients, avoiding the redundancy introduced by reducing high-order constraints, thereby accelerating convergence.

Comparing HONet (Full) against the variant without $\mathcal{L}_{gap}$, Table 3 shows that while the variant achieves high accuracy, the residual gap remains $\sim 10^{-2}$. GCL reduces this gap by two to three orders of magnitude to $\sim 10^{-5}$ and reduces fuzziness to near zero. This indicates that GCL provides an explicit energy-level geometric signal, encouraging the learned objective landscape to favor the target assignment

*Table 3.* Ablation study on architecture variants, loss components, and non-learning baselines under Medium/Hard settings. "Epochs" indicates the epoch at which peak validation accuracy is reached. "Direct Projection" inputs raw clues into the solver; "Biased Projection" adds noise to simulate incorrect priors. HONet follows the inference configuration used in Table 1. Gap and Fuzziness are reported in scientific notation; values below the display threshold are shown as $< 10^{-6}$.

| | | **HONET** | HONET (W/O $\mathcal{L}_{gap}$) | G.T.+CVX | TRANS.+CVX | D. PROJ. | B. PROJ. |
|---|---|---|---|---|---|---|---|
| | GAP | $9.0\times10^{-5}$ | $2.07\times10^{-2}$ | $2.34\times10^{-2}$ | $2.65\times10^{-2}$ | $1.53$ | $2.01$ |
| | FUZZ | $< 10^{-6}$ | $1.12\times10^{-3}$ | $6.46\times10^{-4}$ | $1.47\times10^{-3}$ | $5.32\times10^{-2}$ | $6.84\times10^{-2}$ |
| 26–28 G. | TIME | **11.45MS** | 56.51MS | 20.42MS | 45.24MS | 26.56MS | 35.34MS |
| | EPOCHS | **4** | 8 | 18 | 6 | - | - |
| | ACC. | **100.00%** | 99.80% | 99.80% | 99.80% | 77.90% | 51.70% |
| | GAP | $4.3\times10^{-5}$ | $1.60\times10^{-2}$ | $5.87\times10^{-2}$ | $1.78\times10^{-2}$ | $2.46$ | $2.57$ |
| | FUZZ | $< 10^{-6}$ | $1.07\times10^{-3}$ | $1.37\times10^{-3}$ | $5.92\times10^{-4}$ | $7.14\times10^{-2}$ | $8.85\times10^{-2}$ |
| 23–25 G. | TIME | 54.67MS | 65.81MS | 28.35MS | 36.30MS | 34.25MS | 16.62MS |
| | EPOCHS | **5** | 10 | 17 | 13 | - | - |
| | ACC. | **100.00%** | 99.90% | 99.30% | 99.80% | 56.50% | 36.40% |
| | GAP | $8.4\times10^{-5}$ | $6.72\times10^{-2}$ | $3.98\times10^{-2}$ | $2.89\times10^{-2}$ | $3.77$ | $3.13$ |
| | FUZZ | $3.0\times10^{-6}$ | $4.13\times10^{-3}$ | $2.34\times10^{-3}$ | $1.63\times10^{-3}$ | $9.15\times10^{-2}$ | $1.09\times10^{-1}$ |
| 20–22 G. | TIME | 48.45MS | 76.11MS | 71.31MS | 71.73MS | 14.34MS | 13.77MS |
| | EPOCHS | **7** | 7 | 8 | 9 | - | - |
| | ACC. | **100.00%** | 99.70% | 99.50% | 99.70% | 30.20% | 18.70% |

and align the relaxed solution with the discrete truth.

A counter-intuitive finding is that landscape quality correlates with inference speed. HONet (Full) infers in only 11.45ms, faster than the variant without $\mathcal{L}_{gap}$ (56.51ms). A sharper and better-aligned objective landscape can make the KKT system easier for the QP solver to resolve. This also helps explain the latency increase in OOD scenarios (Section 4.3): slight geometric misalignment may make the relaxed solution harder to align with the target assignment.

Non-learning baselines in Table 3 reveal the limit: under "20–22 Givens," Direct Projection (inputting givens without learning) yields only 30.2% accuracy with a large gap of 3.77. This confirms that without neural guidance, the regularization term can dominate optimization, causing the solution to drift toward fractional interior points. HONet's success in these tasks stems from symbiosis: the fixed QP decoder enforces relaxed feasibility, while the neural encoder shapes the objective landscape.

### 4.5. Versatility Verification: Visual Sudoku and Beyond

This subsection aims to verify the effectiveness of HONet in joint perceptual-reasoning optimization and to probe its boundaries when task-defining inequality constraints are not represented in the fixed exact-cover polytope.

**Visual Sudoku.** To evaluate the framework's compatibility with perceptual inputs, we adopted the Visual Sudoku benchmark (Wang et al., 2019). In this task, the clues are not symbolic numbers but handwritten digit images from the MNIST dataset. This requires the model to jointly optimize visual perception (CNN) and structured decoding modules.

Following the experimental setup of SATNet, baselines include SATNet and SMTLayer (Wang et al., 2023). HONet and SATNet integrate a lightweight LeNet-5 backbone at the front end; SMTLayer attaches an unsupervised auto-encoder pre-trained for 30 epochs before its original framework. All models were trained on a 9k sample set. HONet achieved 98.40% test accuracy, outperforming SATNet (63.20%) and SMTLayer (79.10%). Even more notable is the convergence speed: HONet reached peak performance in 24 epochs, whereas SATNet required 99 epochs. This suggests that structured gradients through the QP layer help the perception module align digit predictions with valid Sudoku completions, leading to faster convergence in the perceptual setting (see Appendix for relevant tables).

**Futoshiki.** To explore HONet's performance boundary when not all constraints are explicitly encoded in the fixed exact-cover formulation, we introduced the Futoshiki task. Futoshiki combines two types of constraints: Exact Constraints, where each row/column must be a permutation of $1 \sim N$ (Latin Square, equality constraints); and Ordinal Constraints, where "Greater than / Less than" relationships exist between adjacent cells (inequality constraints). The ordinal constraints vary across instances and are therefore not included in the shared fixed exact-cover polytope.

We designed a Decoupling Experiment: we retain only the Latin Square constraints within the QP solver, while inequality signs are fed as features to the neural network. We tested on a $5 \times 5$ dataset (15–17 Givens), aiming to examine whether the neural network can implicitly learn inequality logic in the absence of explicit geometric constraints. Under standard sparsity, the model quickly reaches 100% accuracy, suggesting that the equality constraints and ob-

served clues still restrict the candidate space enough for the neural module to learn the remaining ordinal preferences. However, under extreme sparsity with further masked clues, accuracy stagnates at $\sim 85\%$. This phenomenon reveals the importance of explicit geometric constraints: when the QP layer lacks inequality constraints ($\mathbf{Gz} \leq \mathbf{h}$), it constructs only a "Latin Square Polytope," which admits numerous "pseudo-solutions" that violate ordinal rules. Since the solver is blind to these violations and cannot provide constraint-based corrective gradients, a geometric mismatch occurs between neural intuition and solver logic. This contrasts with Sudoku, where the shared exact-cover template provides the global reasoning structure and the neural module only needs to shape the objective for solution selection. This finding supports our core philosophy: known hard logical constraints should be encoded as geometric boundaries when they define the task feasibility, rather than being left entirely to implicit fitting. This also points to a future direction: explicitly integrating instance-specific inequality constraints into the QP formulation, while addressing the resulting optimization and gradient challenges, may help extend fixed-polytope frameworks beyond exact-cover tasks.

## 5. Conclusion and Future Work

In this paper, we proposed the Hypergraph Optimization Network, a neuro-symbolic framework tailored for data-efficient solving of known-constraint Exact Cover tasks. By bridging a structure-preserving Hypergraph Encoder with differentiable fixed-constraint Quadratic Programming, we shift the learning paradigm from implicit statistical fitting to explicit objective parameterization. Crucially, the Geometric Consistency Loss effectively mitigates the Relaxation Gap by encouraging alignment between the relaxed optimum and the target assignment. Empirical results on symbolic and visual Sudoku benchmarks demonstrate that HONet achieves strong data efficiency (100% accuracy with only 9k samples) and strong robustness to sparsity shifts, significantly outperforming purely data-driven baselines.

**Limitations and Future Work.** While HONet performs strongly on tasks with known-constraint exact-cover tasks, scaling HONet presents challenges at both the task and computational levels. At the task level, the fixed-polytope formulation relies on a shared constraint template to provide the global reasoning structure; our Futoshiki experiments reveal the "Geometric Mismatch" that arises when task-defining inequality constraints are not encoded in the fixed solver formulation. At the computational level, differentiable convex optimization layers introduce nontrivial training and inference costs, which may become more pronounced when scaling to larger instances such as $16 \times 16$ Sudoku. Increased sparsity may also affect the stability of objective-landscape shaping. Nevertheless, HONet pro-

vides a division-of-labor perspective. Future work will focus on extending the differentiable solver to explicitly handle instance-specific inequality constraints ($\mathbf{Gz} \leq \mathbf{h}$) and improving the scalability of fixed-polytope decoding, with the goal of extending this framework to broader task families.

## Acknowledgements

This work was supported in part by the National Key R&D Program of China under Grant 2024YFE0200800; in part by the National Natural Science Foundation of China (62471055, U23B2001, 62401080, 62406039, 62321001, 62201072, 62101064); in part by the Fundamental and Interdisciplinary Disciplines Breakthrough Plan of the Ministry of Education of China (JYB2025XDXM107); in part by the High-Quality Development Project of the Ministry of the MIIT (2440STCZB2584); in part by the Fundamental Research Funds for the Central Universities; in part by the Ministry of Education and China Mobile Joint Fund (MCM20200202, MCM20180101); in part by the 2025 Education and Teaching Reform Project Funding at Beijing University of Posts and Telecommunications (2025YZ005).

## Impact Statement

This paper presents work whose goal is to advance the field of Machine Learning, specifically in the domain of Neuro-symbolic optimization. Our proposed framework, HONet, demonstrates significant improvements in data efficiency and convergence speed compared to purely data-driven baselines. This reduction in training data and computational resources aligns with the goals of Green AI, potentially reducing the carbon footprint associated with training neural solvers. Furthermore, by enforcing explicit geometric constraints, our approach enhances the reliability and interpretability of optimization models, which is crucial for deploying AI in safety-critical resource allocation and planning systems. We do not foresee specific negative societal consequences or ethical concerns associated with this work.

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

# A. Proof of Proposition 3.1

*Proof.* We consider the abstract binary exact-cover hypergraph, where each node corresponds to a binary selection coordinate. Let $\mathcal{U}$ be the set of such nodes and $\mathcal{E}$ be the set of hyperedges. For any constraint $c_k$ represented by a hyperedge in $\mathcal{E}$, let $S_k \subseteq \mathcal{U}$ denote its native scope, and define

$$\Psi_k(z_{S_k}) = \begin{cases} \lambda_{\min}, & \sum_{i \in S_k} z_i = 1, \\ \lambda_{\text{penalty}}, & \text{otherwise,} \end{cases} \tag{10}$$

where $\lambda_{\min} < \lambda_{\text{penalty}}$.

**Part I: Local functions on native scopes.** By definition, the local exact-cover function $\Psi_k$ depends only on the count

$$\sum_{i \in S_k} z_i, \tag{11}$$

and is therefore independent of the ordering of variables within $S_k$. Hence, $\Psi_k$ is permutation-invariant on $S_k$.

Consider a permutation-invariant aggregation form

$$g_k(z_{S_k}) = \phi_k\left(\sum_{i \in S_k} \psi(z_i)\right). \tag{12}$$

Let $\psi(z_i) = z_i$, so that

$$\sum_{i \in S_k} \psi(z_i) = \sum_{i \in S_k} z_i. \tag{13}$$

Define

$$\phi_k(t) = \begin{cases} \lambda_{\min}, & t = 1, \\ \lambda_{\text{penalty}}, & t \neq 1. \end{cases} \tag{14}$$

Since $t$ takes values on the finite discrete set $\{0, \ldots, |S_k|\}$, this is an existence statement on the exact-cover assignment domain. Then

$$g_k(z_{S_k}) = \phi_k\left(\sum_{i \in S_k} \psi(z_i)\right) = \Psi_k(z_{S_k}). \tag{15}$$

Therefore, there exists a permutation-invariant hyperedge aggregation form that preserves the native scope $S_k$ and can represent this local exact-cover function.

**Part II: Non-injectivity of clique-expanded pairwise encodings.** We now show that the untyped clique expansion is non-injective with respect to native hyperedge identity.

Consider two hypergraph structures on the same node set $\mathcal{U} = \{1, 2, 3\}$ with hyperedge sets

$$\mathcal{E}_1 = \{\{1, 2, 3\}\}, \qquad \mathcal{E}_2 = \{\{1, 2\}, \{2, 3\}, \{1, 3\}\}. \tag{16}$$

For $(\mathcal{U}, \mathcal{E}_1)$, all three nodes co-occur in one hyperedge, so its untyped clique expansion is the triangle graph $K_3$. For $(\mathcal{U}, \mathcal{E}_2)$, the three binary hyperedges connect the same three pairs, so its untyped clique expansion is also $K_3$. Thus,

$$G_{\text{clq}}(\mathcal{U}, \mathcal{E}_1) = G_{\text{clq}}(\mathcal{U}, \mathcal{E}_2) = K_3. \tag{17}$$

However, the native exact-cover constraints differ. For $(\mathcal{U}, \mathcal{E}_1)$, the feasible assignments satisfy

$$z_1 + z_2 + z_3 = 1, \tag{18}$$

so

$$\mathcal{F}(\mathcal{U}, \mathcal{E}_1) = \{(1, 0, 0), (0, 1, 0), (0, 0, 1)\}. \tag{19}$$

For $(\mathcal{U}, \mathcal{E}_2)$, the constraints are

$$z_1 + z_2 = 1, \qquad z_2 + z_3 = 1, \qquad z_1 + z_3 = 1. \tag{20}$$

Summing them gives

$$2(z_1 + z_2 + z_3) = 3, \tag{21}$$

which has no binary solution. Hence

$$\mathcal{F}(\mathcal{U}, \mathcal{E}_2) = \varnothing. \tag{22}$$

Therefore, two distinct native hyperedge structures with different exact-cover semantics are mapped to the same untyped clique-expanded graph. The map $(\mathcal{U}, \mathcal{E}) \mapsto G_{\mathrm{clq}}(\mathcal{U}, \mathcal{E})$ is non-injective with respect to native hyperedge identity. The two parts together establish the proposition.

$$\square$$

# B. Theoretical Analysis of Geometric Consistency

In this section, we provide the theoretical derivation supporting Geometric Consistency Learning. We analyze how the learned objective coefficients $\mathbf{p}$ interact with the fixed relaxed polytope $\mathcal{P}$ and the QP energy used by the decoder.

## B.1. Regularized Gap and Strong Convexity

Recall the regularized energy optimized by the QP layer:

$$\mathcal{E}_\lambda(\mathbf{z}; \mathbf{p}) = -\mathbf{p}^\top \mathbf{z} + \frac{\lambda}{2} \|\mathbf{z}\|_2^2. \tag{23}$$

For a feasible ground-truth assignment $\mathbf{z}^* \in \mathcal{P}$ and the relaxed solution $\hat{\mathbf{z}}$ obtained from Eq. 7, we define the regularized Relaxation Gap as

$$\mathrm{Gap}_\lambda(\mathbf{z}^*, \hat{\mathbf{z}}; \mathbf{p}) = \mathcal{E}_\lambda(\mathbf{z}^*; \mathbf{p}) - \mathcal{E}_\lambda(\hat{\mathbf{z}}; \mathbf{p}). \tag{24}$$

Since $\hat{\mathbf{z}}$ is the global minimizer of $\mathcal{E}_\lambda(\cdot; \mathbf{p})$ over the convex set $\mathcal{P}$, and $\mathcal{E}_\lambda$ is $\lambda$-strongly convex, the standard strong-convexity inequality (Boyd & Vandenberghe, 2004) gives, for any feasible $\mathbf{z}^* \in \mathcal{P}$,

$$\mathcal{E}_\lambda(\mathbf{z}^*; \mathbf{p}) - \mathcal{E}_\lambda(\hat{\mathbf{z}}; \mathbf{p}) \geq \nabla \mathcal{E}_\lambda(\hat{\mathbf{z}}; \mathbf{p})^\top (\mathbf{z}^* - \hat{\mathbf{z}}) + \frac{\lambda}{2} \|\mathbf{z}^* - \hat{\mathbf{z}}\|_2^2. \tag{25}$$

By first-order optimality of $\hat{\mathbf{z}}$ over $\mathcal{P}$,

$$\nabla \mathcal{E}_\lambda(\hat{\mathbf{z}}; \mathbf{p})^\top (\mathbf{z} - \hat{\mathbf{z}}) \geq 0, \qquad \forall \mathbf{z} \in \mathcal{P}. \tag{26}$$

Taking $\mathbf{z} = \mathbf{z}^*$ yields

$$\mathrm{Gap}_\lambda(\mathbf{z}^*, \hat{\mathbf{z}}; \mathbf{p}) \geq \frac{\lambda}{2} \|\mathbf{z}^* - \hat{\mathbf{z}}\|_2^2. \tag{27}$$

Thus, the regularized energy-gap term complements coordinate-level or task-level supervision by providing an energy-based signal that also controls the $\ell_2$ distance between the relaxed solution and the target assignment. This statement applies to the regularized QP energy $\mathcal{E}_\lambda$, not to the linear term $-\mathbf{p}^\top \mathbf{z}$ alone.

## B.2. Normal-Cone Region of Objective Coefficients

We next characterize which objective coefficients make the target assignment optimal over the fixed polytope.

**Proposition B.1.** *Let* $\mathcal{P} = \{\mathbf{z} \in \mathbb{R}^N \mid \mathbf{Az} = \mathbf{b}, \ \mathbf{z} \geq 0\}$ *be the fixed relaxed feasible polytope, and let* $\mathbf{z}^* \in \mathcal{P}$. *For the QP energy in Eq. 7,* $\mathbf{z}^*$ *is the unique global minimizer over* $\mathcal{P}$ *if and only if*

$$\mathbf{p} - \lambda \mathbf{z}^* \in N_\mathcal{P}(\mathbf{z}^*), \tag{28}$$

*where* $N_\mathcal{P}(\mathbf{z}^*) = \{\mathbf{v} \in \mathbb{R}^N \mid \mathbf{v}^\top (\mathbf{z} - \mathbf{z}^*) \leq 0, \ \forall \mathbf{z} \in \mathcal{P}\}$ *is the normal cone of* $\mathcal{P}$ *at* $\mathbf{z}^*$.

*Proof.* The gradient of the regularized energy is

$$\nabla_{\mathbf{z}}\mathcal{E}_\lambda(\mathbf{z};\mathbf{p}) = -\mathbf{p} + \lambda\mathbf{z}. \tag{29}$$

For a convex differentiable objective minimized over a convex set $\mathcal{P}$, the first-order optimality condition is

$$-\nabla_{\mathbf{z}}\mathcal{E}_\lambda(\mathbf{z}^*;\mathbf{p}) \in N_{\mathcal{P}}(\mathbf{z}^*). \tag{30}$$

Substituting the gradient gives

$$\mathbf{p} - \lambda\mathbf{z}^* \in N_{\mathcal{P}}(\mathbf{z}^*), \tag{31}$$

which proves necessity.

Conversely, suppose Eq. 28 holds. Then, for any $\mathbf{z} \in \mathcal{P}$,

$$(\mathbf{p} - \lambda\mathbf{z}^*)^\top (\mathbf{z} - \mathbf{z}^*) \le 0. \tag{32}$$

Using the quadratic form of $\mathcal{E}_\lambda$, we obtain

$$\mathcal{E}_\lambda(\mathbf{z};\mathbf{p}) - \mathcal{E}_\lambda(\mathbf{z}^*;\mathbf{p}) = (\lambda\mathbf{z}^* - \mathbf{p})^\top (\mathbf{z} - \mathbf{z}^*) + \frac{\lambda}{2}\|\mathbf{z} - \mathbf{z}^*\|_2^2 \tag{33}$$

$$= -(\mathbf{p} - \lambda\mathbf{z}^*)^\top (\mathbf{z} - \mathbf{z}^*) + \frac{\lambda}{2}\|\mathbf{z} - \mathbf{z}^*\|_2^2 \tag{34}$$

$$\ge \frac{\lambda}{2}\|\mathbf{z} - \mathbf{z}^*\|_2^2. \tag{35}$$

Therefore, for every $\mathbf{z} \ne \mathbf{z}^*$ in $\mathcal{P}$,

$$\mathcal{E}_\lambda(\mathbf{z};\mathbf{p}) > \mathcal{E}_\lambda(\mathbf{z}^*;\mathbf{p}), \tag{36}$$

so $\mathbf{z}^*$ is the unique global minimizer. □

**Geometric Interpretation.** The condition $\mathbf{p} - \lambda\mathbf{z}^* \in N_{\mathcal{P}}(\mathbf{z}^*)$ shows that the desired objective coefficients do not correspond to a single target vector. Rather, the shifted normal cone $\lambda\mathbf{z}^* + N_{\mathcal{P}}(\mathbf{z}^*)$ defines a geometric region of coefficients under which the target assignment is optimal for the fixed QP.

This region provides a limited form of tolerance: perturbations of $\mathbf{p}$ that remain inside the shifted normal-cone region do not change the optimality of $\mathbf{z}^*$, while sufficient alignment and scale along favorable directions help counteract the shrinkage induced by the quadratic term.

This perspective explains the role of GCL. If the learned coefficients are too weak or poorly aligned with this normal-cone region, the quadratic regularization may pull the relaxed optimum toward fractional points in the interior of $\mathcal{P}$. By minimizing the regularized energy gap, GCL encourages the learned objective landscape to make the target assignment energetically favorable while remaining consistent with the fixed relaxed feasible region.

**B.3. Role of Auxiliary Warm-Start Supervision**

The normal-cone characterization above shows that the target assignment becomes optimal for the fixed QP when the learned objective coefficients have appropriate direction and scale relative to the shifted normal-cone region of $\mathbf{z}^*$. In sparse-clue settings, the instance-specific evidence for selecting the target completion can be weak, so QP-level losses may provide less direct guidance for shaping the objective landscape. Their gradients are mediated by the relaxed QP solution, whereas the warm-start term acts directly on the pre-QP label-wise potentials.

From this view, warm-start supervision should be interpreted as a practical auxiliary signal rather than a separate feasibility mechanism. It does not explicitly project $\mathbf{p}$ into the shifted normal cone. Instead, it biases the pre-QP label-wise potentials toward the target labels, giving the flattened coefficient vector $\mathbf{p}$ a more favorable initial orientation and scale under sparse observations.

We therefore use an auxiliary warm-start loss on the label-wise potentials before QP decoding. Let $\mathbf{o}_v \in \mathbb{R}^d$ denote the label-wise potential produced for variable node $v$, and let $y_v^*$ denote the corresponding ground-truth label. The warm-start term is

$$\mathcal{L}_{warm} = \frac{1}{|\mathcal{V}|}\sum_{v\in\mathcal{V}} \text{CE}\left(\mathbf{o}_v, y_v^*\right). \tag{37}$$

Equivalently, this is the cross-entropy loss applied to the label-wise potentials at each variable node before they are flattened and temperature-scaled into the QP coefficient vector $\mathbf{p}$.

This term provides direct task-level supervision for the neural potentials, complementing the coordinate-level and energy-level losses. Its role is not to replace the QP decoder, but to provide a practical initialization signal that helps the learned objective landscape favor the target assignment under sparse observations. Together, these losses refine the relaxed solution within the fixed feasible region while keeping the learned objective landscape oriented toward the target assignment.

## C. Constraint Sparsity and Solver Geometry

In this section, we analyze how the set of constraints encoded in the QP solver determines the geometric reasoning structure available to HONet. Here, constraint sparsity refers to missing or unencoded task-defining constraints in the solver formulation, rather than sparse clues in the input.

Let the constraints encoded by the fixed exact-cover decoder define

$$\mathcal{P}_{\text{enc}} = \left\{ \mathbf{z} \in \mathbb{R}^N \mid \mathbf{A}\mathbf{z} = \mathbf{b}, \ \mathbf{z} \geq 0 \right\}. \tag{38}$$

For a task with additional instance-specific constraints, the full relaxed feasible region can be written as

$$\mathcal{P}_{\text{full}}^{(m)} = \left\{ \mathbf{z} \in \mathcal{P}_{\text{enc}} \mid \mathbf{G}^{(m)}\mathbf{z} \leq \mathbf{h}^{(m)} \right\}, \tag{39}$$

where $m$ indexes the problem instance. If these additional constraints are not encoded in the solver, then the decoder optimizes over $\mathcal{P}_{\text{enc}}$ instead of $\mathcal{P}_{\text{full}}^{(m)}$. Since

$$\mathcal{P}_{\text{full}}^{(m)} \subseteq \mathcal{P}_{\text{enc}}, \tag{40}$$

omitting task-defining constraints enlarges the relaxed feasible region and may introduce pseudo-feasible relaxed points:

$$\mathcal{P}_{\text{pseudo}}^{(m)} = \mathcal{P}_{\text{enc}} \setminus \mathcal{P}_{\text{full}}^{(m)}. \tag{41}$$

These points satisfy the constraints known to the solver but violate task rules that were omitted from the solver geometry.

This also affects the normal-cone certificate of optimality. When the full task constraints are encoded, the target assignment is evaluated with respect to

$$N_{\mathcal{P}_{\text{full}}^{(m)}}(\mathbf{z}^*). \tag{42}$$

When only the encoded constraints are used, the solver instead provides the geometry of

$$N_{\mathcal{P}_{\text{enc}}}(\mathbf{z}^*). \tag{43}$$

Thus, missing constraints do not merely change the neural input; they change the feasible geometry over which the QP layer reasons. This provides a geometric explanation for the boundary observed in Futoshiki.

## D. Supplementary Explanation of the Experiment

### D.1. Details of the Experimental Setup

**Datasets.** For the SATNet setting ($\sim 36.2$ Givens), we use the released SATNet Sudoku dataset (Wang et al., 2019). We construct the validation split using the OptNet Sudoku generation script (Amos & Kolter, 2017). For Visual Sudoku experiments, we follow the SATNet MNIST-based formulation and use the corresponding MNIST conversion script to generate the validation split.

For Medium/Hard Sudoku settings (26–28, 23–25, and 20–22 Givens), we sample puzzles from Kaggle's "1 Million Sudoku Games" dataset (Park, 2018). For the Extreme setting (17–19 Givens), we sample puzzles from the RRN dataset (Palm et al., 2018), corresponding to the near-minimal-clue regime for uniquely solvable Sudoku puzzles.

**QP and Inference Configuration.** For the Sudoku settings from $\sim 36.2$ Givens to 20–22 Givens, we use $\lambda = 1$ in the QP decoder. Projection-based baselines directly use observed clues as solver inputs and follow the same $\lambda$ setting and inference scaling as the main experimental configuration for the corresponding difficulty level. To avoid underestimating

these non-learning baselines due to weak input magnitude, we additionally amplify the clue signals by a factor of 10 before projection. The 17–19 Givens setting uses the sparse-regime configuration described in Section 4.3. For the ablation study, each learned variant is evaluated at its best validation checkpoint using the corresponding inference configuration. The full HONet model follows the inference configuration used in the main Sudoku experiments.

**High-Data Baselines.** For high-data baselines, we follow the data sizes, preprocessing protocols, and training settings required by their original implementations. These high-data settings contain the corresponding difficulty distributions used by each baseline, rather than being re-sampled under the 9k-data protocol used for HONet.

### D.2. Supplementary Explanation for Section 4.3

*Table 4.* Zero-shot OOD generalization under the "Single-Distribution Training, All-Distribution Testing" protocol. HONet is trained only on the 20–22 Givens dataset (9k samples) and evaluated on other difficulty levels. For each target difficulty level, inference uses its corresponding $\lambda$ configuration.

| DIFFICULTY | 26–28 G. | 23–25 G. | 17–19 G. |
|---|---|---|---|
| ACC. | 100.00% | 100.00% | 87.50% |
| TIME | 43.11MS | 49.24MS | 52.89MS |

### D.3. Supplementary Explanation for Section 4.4

*Table 5.* Parameter-count comparison for architecture ablations. This table reports the total number of trainable parameters of HONet, Graph Transformer (G.T.), and standard Transformer variants across Sudoku difficulty levels. The backbone and embedding dimensions for G.T. and Transformer are kept identical, allowing the comparison to focus on architectural differences.

| DIFFICULTY | **HONET** | G.T.+CVX | TRANS.+CVX |
|---|---|---|---|
| 26–28 G. | **5,280,521** | 6,343,689 | 6,343,689 |
| 23–25 G. | **7,918,345** | 7,923,209 | 7,923,209 |
| 20–22 G. | **10,556,169** | 14,234,121 | 14,234,121 |

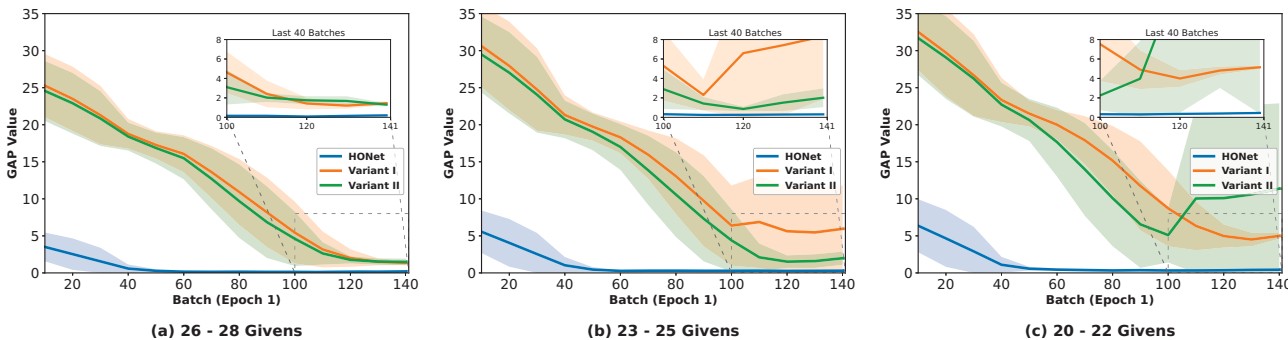

**(a) 26 - 28 Givens**  **(b) 23 - 25 Givens**  **(c) 20 - 22 Givens**

*Figure 4.* Convergence dynamics of different encoder architectures during the first training epoch. The curves illustrate the trajectory of the Relaxation Gap on the training set. Variant I corresponds to the Graph Transformer (based on pairwise attention), and Variant II corresponds to the Global Transformer (structure-agnostic).

### D.4. Supplementary Explanation for Section 4.5

*Table 6.* Performance comparison on Visual Sudoku. Following the experimental setup of SATNet, baselines include SATNet and SMTLayer. HONet and SATNet use a lightweight LeNet-5 backbone at the front end. For SMTLayer, an unsupervised auto-encoder pre-trained for 30 epochs was attached following its original protocol. All models were trained on a 9k sample set.

| METRIC | **HONET (OURS)** | SATNET | SMTLAYER |
|---|---|---|---|
| EPOCHS | **24** | 99 | 30+5 |
| ACC. | **98.40%** | 63.20% | 79.10% |

