# OpenReview forum: "HONet: Data-Efficient Learning for Exact Cover Tasks via Hypergraph Optimization"
_ICML.cc/2026/Conference — ICML 2026 regular_

### Official Review · Reviewer_4MdD · 2026-03-06

**Soundness:** 3
**Presentation:** 2
**Significance:** 3
**Originality:** 3
**Overall Recommendation:** 4
**Confidence:** 4

**Summary:**

The paper introduces HONet, a neuro-symbolic framework designed for the data-efficient solving of Exact Cover Problems. It integrates a topologically complete Deep Residual Hypergraph Encoder with a differentiable Equality-Constrained Quadratic Programming (EQP) layer to shift the learning paradigm from implicit statistical fitting to explicit rule parameterization. By relaxing discrete constraints into a convex polytope and utilizing a novel Geometric Consistency Loss (GCL) theoretically grounded in KKT complementarity, the model reshapes the objective energy landscape to align the continuous global minimum with the valid discrete vertex. Empirical results demonstrate that HONet achieves 100% accuracy on $9\times9$ Sudoku with only 9k training samples, significantly outperforming baselines that require orders-of-magnitude more data, while maintaining robust zero-shot generalization in extremely sparse settings (17-19 Givens).

**Compliance With Llm Reviewing Policy:**

Affirmed.

**Final Justification:**

After reviewing the authors' rebuttal, I find that my concerns have been largely addressed through additional experimental results and clarifications. These responses improve the clarity of the paper and better justify the design choices.

However, my overall assessment remains unchanged. While the rebuttal is helpful, I do not find that it substantially strengthens the overall contribution to the extent of warranting a higher score. Therefore, I maintain my original rating.

**Key Questions For Authors:**

1. Explicitly define the mathematical mapping between the flattened vector $\mathbf{z}$ and the matrix notation $Z_{i,j}$ used in Equation (1). Commit to revising the manuscript to utilize standard bolding conventions for vectors ($\mathbf{z}$, $\mathbf{p}$, $\mathbf{b}$) and matrices ($\mathbf{A}$, $\mathbf{Z}$) to resolve current ambiguities. (Addressing this is critical for the Presentation score).
2. Section 3.1 claims to address "general CSPs" but strictly formulates the domain $\Omega$ using linear equalities ($\mathbf{A}\mathbf{z} = \mathbf{b}$). How does the proposed formulation accommodate non-linear constraints or inequality constraints (e.g., the "All-Different" constraint)? Please clarify or appropriately narrow the scope of your claims.
3. Please correct the misleading statement at the end of Section 3.1 that claims the problem has been transformed into a "continuous geometric optimization problem". The domain $\Omega$ in Equation (1) is explicitly defined over $\{0,1\}^N$, making it a discrete optimization problem at that specific point in the text.
4. Differentiating through KKT conditions via CvxpyLayers typically introduces significant computational bottlenecks. While the paper highlights polynomial time complexity and provides inference latencies, what is the wall-clock training time per epoch? How does this overhead scale when transitioning to larger grids (e.g., $16 \times 16$ Sudoku)?
5. In the theoretical sparsity limit experiments (17-19 Givens), the standard regularization setting ($\lambda=1$) fails, requiring manual attenuation of $\lambda$. How sensitive is the "Fixed Polytope" paradigm to $\lambda$ in general OOD scenarios, and can this parameter be learned or dynamically scheduled to avoid manual tuning?
6. Table 3 clearly demonstrates the necessity of the GCL loss. However, since the paper states that \mathcal{L}_{warm} provides a "massive initial driving force", it is difficult to disentangle its exact contribution from \mathcal{L}_{gap}. Providing a plot of the loss components (\mathcal{L}_{gap}, \mathcal{L}_{MSE}, \mathcal{L}_{warm}) over the training epochs would greatly clarify their individual roles across different training phases.

**Limitations:**

Yes

**Strengths And Weaknesses:**

Strengths: The paper tackles a problem in neuro-symbolic AI—data inefficiency under hard constraints—with an original and elegant approach. By combining a Deep Residual Hypergraph Encoder with a differentiable EQP layer, the framework strictly enforces structural priors without information loss. The empirical soundness is exceptional: HONet achieves 100% accuracy on the $9\times9$ Sudoku task using only 9k samples, vastly outperforming data-heavy baselines like RRN and SATNet. Additionally, the Geometric Consistency Loss (GCL) provides a theoretically sound, KKT-grounded mechanism to successfully bridge the continuous-to-discrete "Relaxation Gap".

Weaknesses: The presentation and theoretical soundness of the mathematical formulation are severely flawed. For example, Section 3.1 claims to define a "general CSP" but strictly limits constraints to linear equalities ($\mathbf{A}\mathbf{z} = \mathbf{b}$), failing to account for how non-linear or inequality constraints would be handled. The text mathematically contradicts itself by stating this formulation transforms the problem into a "continuous" optimization, even though the feasible domain $\Omega$ remains explicitly discrete ($\{0, 1\}^N$) at that stage. The notation is unacceptably imprecise for a top-tier venue: the mapping between the flattened vector $\mathbf{z}$ and the double-index matrix $Z_{i,j}$ is entirely undefined, and standard bolding conventions for vectors and matrices are completely ignored. These foundational errors make the core methodology confusing and require major revision.

---

> ### Author Rebuttal · Authors · 2026-03-31
>
> Thank you for the careful review. We appreciate that you found the core idea original and elegant, and that you viewed the empirical and geometric contributions as strong. Added figures and tables are available at anonymous URL https://anonymous.4open.science/r/rebuttal33745/. Below we answer your key questions directly.
>
> 1. On the mapping between the flattened vector and the indexed assignment notation, this definition should be made more explicit. Our intended construction is to define an assignment matrix $\mathbf Z$ and flatten it through $\mathbf z=\operatorname{vec}(\mathbf Z)$ into a binary indicator vector $\mathbf z$. In revision we will make this mapping explicit and adopt standard bold notation for vectors ($\mathbf z,\mathbf p,\mathbf b$) and matrices ($\mathbf A,\mathbf Z$), to remove ambiguity and improve presentation.
>
> 2. On the scope of Section 3.1, our original intent was to motivate the modeling idea from a broader CSP perspective, but the current wording overstates the claim. Section 3.1 actually gives a fixed-feasible-region, equality-dominant structured-assignment formulation consistent with the exact-cover-style tasks validated here. Nonlinear, general-inequality, and broader All-Different constraints should therefore indeed not be read as directly covered; in this paper, non-equality constraints appear mainly in the boundary-analysis setting. In revision we will narrow this scope claim and present broader constraint types as extensions.
>
> 3. This point also needs clarification. At Eq. (1), the feasible set is still discrete ({$0,1$}$^N$); the problem only becomes a continuous geometric optimization after the convex relaxation introduced later in Section 3.3. We will revise the wording explicitly.
>
> 4. On computational bottlenecks, under matched 1000-instance training settings, 9x9 Sudoku takes about 9-11 minutes per epoch to train, while 16x16 takes about 72-74 minutes. As you note, and especially as grid size grows, the main overhead comes from differentiation through the KKT system. **Table 7** at the anonymous URL reports variable count, constraint count, inference latency, and memory, making this scaling profile more explicit. We also ran a preliminary 9x9 version with a first-order FFO layer. Current observations suggest some wall-clock potential, but its optimization behavior is still clearly weaker than the CvxpyLayer-based version, with less satisfactory loss descent. We therefore view it as a promising acceleration direction rather than a mature replacement, and will discuss it more clearly in future work.
>
> 5. On $\lambda$ sensitivity, the reduced $\lambda$ used in the 17-19 givens setting addresses numerical stability in extreme-sparsity training, rather than a mandatory manual retuning step for OOD inference. As discussed in Appendix C, shrinking normal cones and stability radii under extreme sparsity make a smaller $\lambda$ helpful for entering an effective descent regime more quickly, whereas in denser settings too small a $\lambda$ can make the landscape overly sharp and destabilize training. We also compared different $\lambda$ values in the original 20-22 givens $\rightarrow$ 17-19 givens zero-shot OOD evaluation. We found some sensitivity, but not extreme fragility: $\lambda=1.0/0.5/0.1$ gives 87.2%/88.5%/87.5% puzzle accuracy, with inference times of $0.0736\text{s}/0.1414\text{s}/0.2334\text{s}$, respectively. Thus a moderate $\lambda$ is slightly better, but the default $\lambda=1$ does not fail outright, while smaller $\lambda$ increases cost. We therefore view this as a practical limitation of the current differentiable QP instantiation under extreme sparsity, rather than evidence that the fixed-polytope paradigm generally requires manual retuning in OOD inference. Dynamic scheduling of $\lambda$ is a natural direction and also aligns better with the accuracy-efficiency tradeoff observed here; fully learnable $\lambda$ would likely require additional stabilization.
>
> 6. On disentangling $\mathcal{L}\_{\mathrm{warm}}$ and $\mathcal{L}\_{\mathrm{gap}}$, we added epoch-level loss trajectories for the 20-22 and 17-19 givens settings together with a 17-19 loss ablation; see **Figure 5** and **Table 8** at the anonymous URL. **Figure 5** shows that in 20-22 givens, weighted $\mathcal{L}\_{\mathrm{gap}}$ stays higher and dominates the early-stage decrease, while weighted $\mathcal{L}\_{\mathrm{MSE}}$ remains smaller and smoother; in 17-19 givens, $\mathcal{L}\_{\mathrm{warm}}$ acts mainly in the early stage. Consistently, **Table 8** shows that the full objective gives 91.80% puzzle accuracy, while removing $\mathcal{L}\_{\mathrm{warm}}$ drops it to 78.30%; removing the other two gives 87.10% and 88.20%. We will use these additions to state the division of labor among the three losses more clearly.
>
> We appreciate your suggestions, and hope these revisions address your concerns about scope, presentation, and scalability more clearly.

---

> > ### Author Rebuttal · Reviewer_4MdD · 2026-04-03
> >
> > I thank the authors for their thorough and highly constructive rebuttal. The additional data (Tables 7 & 8, Figure 5) and the commitments to fix the mathematical notation and narrow the scope claims effectively address my primary concerns regarding presentation and clarity. The inherent computational overhead of the KKT layer and the restricted scope of the methodology remain practical limitations.

---

> > > ### Author Response · Authors · 2026-04-07
> > >
> > > Thank you for the thoughtful follow-up and for indicating that our rebuttal has addressed your primary concerns. We are glad that the additional results and our clarifications on notation, scope, and the discrete/continuous distinction were helpful. We will make these changes explicit in the revised manuscript and further clarify the scope and positioning of the framework. We sincerely appreciate your constructive feedback.

---

### Official Review · Reviewer_5GFc · 2026-03-09

**Soundness:** 3
**Presentation:** 2
**Significance:** 3
**Originality:** 3
**Overall Recommendation:** 4
**Confidence:** 3

**Summary:**

This paper proposed a hypergraph optimization network for solving the exact cover problem. The constraints are modeled through a hypergraph neural network (HGNN). Differentiable equality-constrained quadratic programming is utilized to shift the learning paradigm from implicit statistical fitting to explicit objective parameterization. The empirical results can demonstrate that the proposed HONet is efficient and exceptionally robust to sparsity shifts. The limitation of the proposed method is well discussed in the conclusion section.

**Compliance With Llm Reviewing Policy:**

Affirmed.

**Final Justification:**

I have read the rebuttal, and my overall assessment and score are unchanged.

**Key Questions For Authors:**

Please see Weaknesses.

**Limitations:**

Yes

**Strengths And Weaknesses:**

**Strengths**
1. The proposed HONet can reaches 100% accuracy with only 9k training samples and maintains robustness even when the training set is reduced to 1%.
2. Theoretical analysis of geometric consistency is illustrated with a sound theoretical foundation.


**Weaknesses**
1. The discussion about the complexity and training stability is missing. It is unclear how to design the loss weighting coefficients.
2. Proposition 3.1 claims that hyperedge convolution provides a “lossless mapping”. The definition of lossless mapping is not clear. A more detailed comparison with related GNNs or Topological Neural Networks (TNNs) is needed. At least it should be included in the Related Work section.
3. Minor concerns: There are some typos in the manuscript. For example, “Deep Learnin” in Line 47.

---

> ### Author Rebuttal · Authors · 2026-03-31
>
> Thank you for the positive assessment. We are glad that you found the data-efficiency result compelling and the geometric-consistency analysis theoretically grounded. Below we respond to your three concerns in turn.
>
> 1. On complexity, training stability, and loss weights, we add the following clarification. Our intended design is that the three loss terms play different roles: $\mathcal{L}\_{\mathrm{gap}}$ imposes a geometric constraint on the energy gap between the discrete vertex $\mathbf z^\*$ and the relaxed solution $\hat{\mathbf z}$, explicitly reshaping $\mathbf p$ until $\mathbf z^\*$ aligns as the global energy minimizer within the relaxed polytope; $\mathcal{L}\_{\mathrm{MSE}}$ provides a direct geometric supervision anchor; and $\mathcal{L}\_{\mathrm{warm}}$ is introduced only in the extreme-sparsity regime as an explicit warm-start signal. To address your concerns about training stability and loss weighting, we added weighted loss trajectories for the 20-22 and 17-19 givens settings, together with a loss ablation at the 17-19-givens sparsity limit; see **Figure 5** and **Table 8** at anonymous URL https://anonymous.4open.science/r/rebuttal33745/. The trajectories show that in the 20-22-givens setting, weighted $\mathcal{L}\_{\mathrm{gap}}$ remains at a higher magnitude and dominates the early-stage decrease, while weighted $\mathcal{L}\_{\mathrm{MSE}}$ stays smaller and smoother; in the 17-19-givens setting, $\mathcal{L}\_{\mathrm{warm}}$ acts mainly in the early stage. Consistently, the ablation shows that the three losses are not redundant: the full objective reaches 91.80% puzzle accuracy, removing $\mathcal{L}\_{\mathrm{warm}}$ drops it to 78.30%, and removing $\mathcal{L}\_{\mathrm{gap}}$ or $\mathcal{L}\_{\mathrm{MSE}}$ gives 87.10% and 88.20%, respectively. To address your concern about complexity, we also added an inference-side complexity table for 9x9 and 16x16 Sudoku; see **Table 7** at the same anonymous URL. It reports variable count, constraint count, inference latency, and CPU / GPU memory, and makes the current scaling characteristics more explicit. Since the current CvxpyLayers-based implementation solves a constrained strictly convex QP in the forward pass and performs implicit differentiation through the KKT system in the backward pass, both time and memory become more sensitive to the number of variables, the number of constraints, and the constraint density. We will use these additions to explain the motivation for the loss weights and to discuss training stability and complexity more explicitly in the revision.
>
> 2. On Proposition 3.1, your suggestion is very helpful. Our intended point is mainly a contrast with clique-expanded pairwise encodings, not a universal impossibility claim for all graph-based models. More precisely, by “lossy” we mean structural information loss: when native higher-order constraints are reduced through clique expansion into a pairwise graph representation, distinct higher-order templates may collapse into the same pairwise structure. The appendix examples of all-zero ambiguity and structural indistinguishability were meant to illustrate exactly this point. Correspondingly, by “lossless” we do not mean a strict engineering guarantee, but rather whether the representation still explicitly preserves the native higher-order constraint template and its identity. In the revision, we will more clearly position Proposition 3.1, clarify what we mean by “lossless / lossy”, and specify the intended comparator scope, while shifting the emphasis toward structural faithfulness to native higher-order constraint templates and adding more targeted comparison in Related Work with clique-expanded GNNs, structure-preserving graph formulations, and broader TNN / higher-order topological models.
>
> 3. On presentation, thank you for pointing out the typo (“Deep Learnin”) and similar issues. We will carefully proofread the manuscript and fix notation, formatting, and typographical problems throughout.
>
> Overall, we hope the above additions and clarifications more clearly address your concerns about complexity, Proposition 3.1, and presentation.

---

> > ### Author Rebuttal · Reviewer_5GFc · 2026-04-03
> >
> > Thanks for the detailed response, I appreciate it.
> >
> > The loss is formulated as a weighted sum of three terms, and each term is given an intuitive geometric role. However, the overall objective still appears largely heuristic, and the paper does not clearly justify how the weighting coefficients are chosen.

---

> > > ### Author Response · Authors · 2026-04-08
> > >
> > > Thank you for your further response. In our previous reply, we mainly explained the geometric roles of the three loss terms, but we did not explain the rule for choosing the weighting coefficients clearly enough. Therefore, what we would like to further explain here is that the coefficients are set according to a scale-balancing rule, whose goal is to allow the three loss terms to play their intended roles at appropriate relative magnitudes and time scales under different sparsity settings. We will make this point clearer in the revised manuscript.
> > >
> > > We use $\mathcal{L}\_{\mathrm{MSE}} $ as the reference anchor term and fix its coefficient to $1$. This is because it directly constrains the relaxed solution $\hat z$ toward the discrete ground truth $z^*$ and provides the most stable and direct supervision signal in the early stage of training. Correspondingly, the coefficient of $\mathcal{L}\_{\mathrm{gap}}$ is set according to the principle that, after weighting, it should remain at a comparable scale to the anchor term without overly suppressing it. More specifically, we first examine the unweighted numerical scales of the different loss terms, and then choose an appropriate coefficient so that $\mathcal{L}\_{\mathrm{gap}}$ can continuously impose a geometric constraint during training, explicitly reshaping $p$ without harming the overall optimization stability. For $\mathcal{L}\_{\mathrm{warm}}$, we only activate it under the extremely sparse 17-19 givens setting and keep its weight small, because its role is mainly to provide an explicit warm-start signal in the early stage of training, rather than to dominate the optimization throughout the entire training process.
> > >
> > > Figure 5 further illustrates how the default coefficients calibrate the different loss terms to appropriate time scales and relative magnitudes. Under the 20-22 givens setting, the weighted $\mathcal{L}\_{\mathrm{gap}}$ and $\mathcal{L}\_{\mathrm{MSE}} $ both decrease rapidly in the early stage of training and soon enter a low and stable magnitude range, without one term persistently dominating the other. This suggests that, in this setting, the coefficients mainly serve a scale-calibration role, allowing the geometric correction signal to remain present while maintaining stable training. By contrast, under the more difficult 17-19 givens setting, $\mathcal{L}\_{\mathrm{warm}}$ mainly acts in the early stage of training and then quickly becomes secondary; meanwhile, $\mathcal{L}\_{\mathrm{gap}}$ remains at a relatively higher magnitude over a longer training interval as the main geometric correction signal, while $\mathcal{L}\_{\mathrm{MSE}} $ decreases earlier to a lower and smoother level and continues to provide a direct geometric supervision anchor. We therefore view these coefficients more as an adaptation of the optimization pressure across different sparsity settings.
> > >
> > > We will further clarify in the revised manuscript that our goal is not to claim theoretical optimality of these coefficients, but rather to explain that they are set according to a clear scale-balancing rule to support stable training under different sparsity settings.
> > >
> > > Again, we sincerely thank you for your response and reminder.

---

### Official Review · Reviewer_Mgxc · 2026-03-13

**Soundness:** 3
**Presentation:** 2
**Significance:** 3
**Originality:** 3
**Overall Recommendation:** 4
**Confidence:** 3

**Summary:**

This paper proposes HONet, a neuro-symbolic framework for solving exact-cover style constraint satisfaction problems. The main target task is Sudoku, treated as a structured assignment problem where solutions must satisfy strict logical constraints. For evaluation, the authors also extended their framework to Visual Sudoku and Futoshiki to explore its extensibility. HONet uses a neural network to learn objective coefficients and a differentiable equality-constrained quadratic programming layer that integrates models (constraints) directly.

**Compliance With Llm Reviewing Policy:**

Affirmed.

**Key Questions For Authors:**

Please refer to the questions in the weaknesses.

**Limitations:**

yes

**Strengths And Weaknesses:**

Strength

This paper is well motivated and consistently designed. The main technique is using constraint-oriented optimization. Concretely, the model fixes the feasible region via symbolic constraints and introduces Geometric Consistency Loss (GCL) to specifically address the claimed relaxation-gap issue. From design, experiments to theories, they are generally consistent.

Weakness
1. The method appears specialized to equality-constrained problems with known symbolic structure. A major part of the performance advantage seems to come from fixing the feasible region. The Futoshiki results also show that once inequality constraints matter, the current solver formulation is no longer sufficient. Is it possible to extend this framework to some non-equality-constrained problems? It would be stronger if the authors could provide concrete examples, as they have for exact-cover-type problems.

2. HONet used a differentiable QP layer. In the experiments, instances are limited to relatively small structured puzzles. The paper reports inference times, but does not show how computation scales with larger instances or denser constraint systems. Could the authors give more analysis regarding these aspects?

---

> ### Author Rebuttal · Authors · 2026-03-31
>
> Thank you for the constructive review. We are glad that you found the paper internally consistent from design to experiments and theory. Below we respond to your two concerns in turn.
>
> 1. Your reading is correct: the current submission is primarily targeted at a setting where the feasible region is fixed by known symbolic constraints, and the present implementation is centered on equality-dominant exact-cover-style tasks.
> - Regarding Futoshiki, we would like to clarify one point that is easy to misunderstand but important for positioning the paper. The core modeling choice in HONet is that the optimization layer enforces a fixed symbolic structure, while the neural network learns objective coefficients over that fixed feasible region. The main difficulty in Futoshiki is therefore not merely the presence of inequalities. Relative to Sudoku, the crucial difference is that the ordinal constraints are not only part of the task logic, but also vary across instances. Accordingly, in the current Futoshiki experiment, these ordinal constraints were not fully encoded into the optimization layer; instead, we adopted a setting in which the solver captures only the fixed equality structure while the front-end network absorbs the remaining relational logic. Strictly speaking, this setting no longer fully satisfies our core modeling condition that all task-defining constraints are fixed and encoded in the solver. This setup lets us probe the boundary of the fixed-feasible-region division of labor, and it is precisely under this boundary case that we observe the geometric mismatch reported in the paper.
> - Conceptually, the framework does not inherently exclude linear inequality constraints: if such constraints are known and fixed, they can in principle be incorporated into the differentiable optimization layer. As a preliminary probe in this direction, we ran a small fixed-inequality pilot experiment on a Futoshiki subset, where the inequality template was fixed and explicitly added to the optimization layer. We did observe signs of improvement, but the gains are not yet stable enough across settings to elevate this into a new main empirical claim in the rebuttal. A more appropriate interpretation is that mixed-constraint extension is promising, but still requires more systematic validation. In the revision, we will add a clearer discussion of this pilot mixed-constraint extension in the Futoshiki section and distinguish it explicitly from the boundary analysis already reported in the main text, so as to address your concern more directly.
>
> 2. Your point about scalability is important. The computational cost of the current framework can be separated into two parts: the HGNN encoder, whose cost grows mainly with the incidence structure and network depth, and the differentiable QP layer, which is the part of the current implementation that most merits separate computational analysis. More concretely, in the current CvxpyLayers-based implementation, the optimization layer solves a constrained strictly convex QP at each forward pass and performs implicit differentiation through the KKT system during backpropagation. As a result, both time and memory become more sensitive to the number of variables, the number of constraints, and the constraint density. This is also where the KKT-based differentiable solver mechanism becomes most relevant in practice. At the same time, this layer is also the key source of HONet's strict feasibility enforcement and effective geometric gradients under fixed symbolic structure. Concretely, when moving from 9x9 to 16x16, the optimization dimension grows from 729 to 4096 and the raw equality-constraint count from 324 to 1024. To address this more concretely, we added **Table 7**  at anonymous URL https://anonymous.4open.science/r/rebuttal33745/, comparing 9x9 and 16x16 in terms of variable count / constraint count / inference latency / inference memory footprint. The table makes the overall scaling trend of the current implementation more explicit and helps explain the corresponding increase in latency and memory. We are also actively exploring sparse linear algebra implementations and differentiable first-order optimization layers that may be better suited to this setting. The former may further reduce large-scale solve cost, while the latter may offer better wall-clock efficiency but also bring weaker or less stable gradient signals. We will discuss this direction more explicitly in the future-work section of the revision.
>
> Thank you again for the constructive suggestions. We hope these clarifications address your concerns about the paper's scope, boundary of applicability, and scalability more clearly.

---

> > ### Author Rebuttal · Reviewer_Mgxc · 2026-04-03
> >
> > Thank you very much for the reply. Even though I will keep the current score, I still have significant concerns about the algorithms' ability to address other problems, and I am not sure how the paper will have a broad influence on the ML community.

---

> > > ### Author Response · Authors · 2026-04-07
> > >
> > > Thank you very much for your further clarification. We understand your concern about the paper’s broader influence, and we believe this is a very important reminder. It also helps us more accurately clarify the positioning of the paper and better calibrate its claims.
> > >
> > > We would also like to clarify more explicitly that HONet is mainly focused on a setting where hard symbolic constraints are already known and shared across instances. Rather than pursuing broader problem coverage, the goal of this paper is to characterize, in this setting, what we believe is a modeling question worth studying in its own right: when the feasible region is already determined by known symbolic structure, how should learning and optimization divide their roles?
> > >
> > > From this perspective, we hope the contribution of the paper can be understood as a clearer, reusable modeling perspective, rather than as a task-specific engineering design for a particular puzzle. More concretely, what we hope to highlight is the following: when the hard constraint template is already known, relying primarily on statistical fitting often forces the network to take on both perception and rule re-discovery, which can lead to high sample complexity; once continuous relaxation is introduced to obtain differentiable training signals, it naturally creates a mismatch between the relaxed solution and the valid discrete solution; and when the constraints themselves have native higher-order structure, compressing them into pairwise relations at the representation level may also result in structural information loss.
> > >
> > > Therefore, what we would like to emphasize is that the paper is not intended to claim that it already covers a broader family of problems. Rather, through exact-cover-style tasks as a relatively clean testbed, we aim to clearly demonstrate a modeling division of labor: while preserving the fidelity of the constraint structure as much as possible, the optimization layer explicitly carries the fixed feasible region, the neural network learns the objective parameters, and geometric alignment is used so that continuous training signals can be mapped more stably to discrete feasible solutions.
> > >
> > > In this sense, we understand the paper’s broader influence as coming more from the fact that, in an important and clearly defined structured setting, it helps clarify how learning and optimization should divide their roles, and also provides a clear starting point for learnable structured solving. We also fully agree that extending the framework to mixed constraints or instance-varying constraints is a very important future direction; however, we view this more as a follow-up extension of the current work, rather than as a claim that has already been fully established in this paper. In the revision, we will further clarify the scope boundary of the paper and communicate the intended positioning of our work more accurately.
> > >
> > > Again, we sincerely thank you for your further comments.

---

### Official Review · Reviewer_JWnB · 2026-03-17

**Soundness:** 2
**Presentation:** 1
**Significance:** 2
**Originality:** 2
**Overall Recommendation:** 3
**Confidence:** 3

**Summary:**

The paper proposes a hypergraph approach to solving the exact cover problem. The paper generates a continuous parametrized relaxation of the problem that uses confidence scores generated by a neural net as parameters and a differentiable convex optimization solver to solve a convex regularized version of the problem in an end to end fashion. The neural net produces the scores through message passing steps that resemble belief propagation updates. The paper introduces additional losses that force the relaxed solutions to have similar or larger energy than the integral solutions, thus encouraging the integral solutions to become global minimizers of the problem. The method is deployed in sudoku puzzles where it shows strong results and superior data efficiency.

**Compliance With Llm Reviewing Policy:**

Affirmed.

**Final Justification:**

The authors have addressed some of my comments but I have reservations regarding the overall scope of the paper and the potential impact of the contribution as there seems to be a rather narrow focus on certain kinds of exact cover problems. I certainly think there's merit in the overall methodology that the paper follows and the techniques employed are promising, but to be more convinced I would like to see more comprehensive evaluation of the method across different kinds of instances and larger scales. Currently I am reluctant to recommend acceptance although I wouldn't be opposed to it if the AC decided to accept this.

**Key Questions For Authors:**

- What are the overall computational costs of this method and how would they scale with the size of the instance. I feel like this is not adequately discussed in the paper. The method looks quite fast on sudoku but suppose the number of items and number of constraints grew, what would the inference time and memory costs look like? A study/comparison against classical algorithms would help clarify how the method stacks up against alternatives when accounting for various budget constraints.

**Limitations:**

Yes

**Strengths And Weaknesses:**

### Strengths

- Interesting reparametrization of the problem that naturally introduces a neural network architecture in the optimization pipeline.
- Sensible use of a parametrized convex relaxation for the problem.
- Promising empirical results, in particular in terms of data efficiency. The method performs consistently well and is fast on all the task that have been tested.
- The direction of decoupling the constraint structure from the objective and focusing on shaping the objective with the neural net is certainly worth exploring more and the paper does a good job of showing that it has potential.

### Weaknesses
- Writing: There are several issues with the writing here that impact the overall presentation of the work. Equation 2 for example, did you mean to write something like  $ \text{argmin}_{\mathbf{z}} - \mathbf{z}^\top \mathbf{p} $ on the right hand side? Shouldn't the polytope around line 188 include the simplex constraints (sum to 1) and/or the variable constraint $\mathbf{z} \leq 1$? As another example the proof of proposition 3.1 also has a bunch of formatting issues so it's kind of hard to read as well.
- I find the paper's discussion of the state of the art and best known methods for the exact cover problem insufficient. Since this is a specialized solver for the problem, how would it do in vehicle routing and partitioning instances (e.g., from [3]). How would it compare against classical algorithms (again like the one in [3]). I appreciate that the method seems to work on sudoku but this is also a fairly structured and clean problem setting. I would like to get a better understanding of what to expect if we were to deploy this method on other challenging combinatorial instances of exact cover. Note that I am not intimately familiar with the literature on this problem so this is one more reason why I feel like the paper should do a better job of covering related  work since it would help readers get a better understanding of the significance of the contribution.

-  The paper also quite cleanly connects to not only 'predict then optimize' but also to the 'blackbox autodifferentiation' techniques (see [1,2]) and those pursue fairly similar ideas so the connections to those techniques merits further discussion. The key distinction seems to be that in the paper here the authors fix the constraints while in related work the constraints are jointly learned.

- Proposition 3.1 and its proof significantly lack detail so it is rather hard to evaluate the claim that is being made. What exactly is meant by lossy, what model class for GNNs is considered and what representation of the problem to establish this kind of claim is handwaved and not discussed in appropriate detail. Couldn't a GNN on a variable-constraint graph for example overcome the issues suggested here and I also don't see why universal function approximation on sets is quite necessary for the 'lossless map". Furthermore the universal representation theorem on sets provides a condition on the dimensionality of the embeddings. Even if I grant that it is needed, do you follow that in practice and fix your embedding dimension to be sufficiently large? Maybe I'm misunderstanding something but I would argue it's hard not to because the writing is too vague there.

- More ablations on the loss (e.g., on $\mathcal{L}_\text{warm}$) would be nice. The paper provides a plausible explanation for it in the appendix but some experiments to demonstrate its utility would help.

Overall, I think this is an interesting paper but I have pointed out several issues which prevent me from recommending to accept it. However, I am definitely open to reconsidering my evaluation if the authors address the issues I brought up.

1. Paulus, Anselm, Georg Martius, and Vít Musil. "LPGD: A general framework for backpropagation through embedded optimization layers." arXiv preprint arXiv:2407.05920 (2024).
2. Vlastelica, Marin, et al. "Differentiation of blackbox combinatorial solvers." arXiv preprint arXiv:1912.02175 (2019).
3. Nishino, Masaaki, Norihito Yasuda, and Kengo Nakamura. "Compressing Exact Cover Problems with Zero-suppressed Binary Decision Diagrams." IJCAI. 2021.

---

> ### Author Rebuttal · Authors · 2026-03-31
>
> Thank you for your detailed review. The draft could clarify its scope, math presentation and scalability.
> 1. On mathematical clarity: in Eq. (2), we mean that the discrete ground-truth assignment minimizes linear energy over the discrete feasible set, i.e., $\mathbf z^*=\arg\min_{\mathbf z\in\Omega}-\mathbf p^\top\mathbf z$. Near line 188, simplex and other hard structural constraints are jointly encoded in $\mathbf A\mathbf z=\mathbf b$; under this form, $\mathbf z\le1$ is implied by simplex constraints and nonnegativity and need not be written separately. We will revise the math presentation accordingly to avoid ambiguity, and fix the formatting issues in Proposition 3.1 and its proof.
> 2. Your exact-cover positioning comments are important.
> - The paper focuses on how, over task families with shared constraint templates, effective solving can be achieved more data-efficiently than with other learnable methods; this is an important question in learnable exact-cover-style solving, which is why Sudoku / Visual Sudoku are the main empirical platforms. In this setting, HONet's core modeling choice is to let the network learn objective parameters over the fixed polytope induced by known symbolic constraints, thereby shaping the energy landscape. Correspondingly, the classical exact-cover search / compression line represented by [3] handles a symbolic exact-cover setting in which candidate sets and constraint structures vary across instances, whereas the paper studies a more learnable subsetting of exact-cover-style tasks, namely the fixed-feasible-region setting.
> - For broader-instance or non-equality concerns, we use Futoshiki as a boundary-analysis case for the fixed-feasible-region setting and HONet's division-of-labor boundary. Its key difficulty is not just inequalities, but that some task-defining constraints are not fully encoded in the solver. To probe this concern, we also ran a small fixed-inequality / mixed-constraint pilot and saw some improvement, but not yet enough for a new main claim. Relative to [1] and [2], the paper is also in the line of upstream training via downstream optimization, but its focus is not only through-solver differentiation; it emphasizes a fixed-feasible-region + objective shaping + geometric alignment paradigm for exact-cover-style neuro-symbolic solving and the data-efficiency gain this brings under known hard symbolic constraints. In revision, we will clarify the paper's relation to the line represented by [3] and the line represented by [1]-[2], and sharpen its scope.
> 3. Regarding Proposition 3.1, our intended contrast is with clique-expanded pairwise encodings, not a claim that all graph-based models fail. By “lossy” we mean structural information loss: when native high-order constraints are reduced through clique expansion into a pairwise graph representation, distinct high-order templates can collapse into the same pairwise structure, as illustrated by the appendix examples of all-zero ambiguity and structural indistinguishability. You also note that the variable-constraint graph is a stronger comparator, and rightly so; since HONet preserves the native variable-constraint incidence structure in hypergraph form, our proposition is mainly aimed at clique-expanded pairwise encodings that do not preserve native high-order constraint identity. The original intent of the set-function universality citation was existence-style motivation, not a strict practical “lossless” claim, nor does it mean that a sufficiently large hidden dimension makes it automatic. In the revision, we will clarify the wording of Proposition 3.1 and delimit its representational premise and comparator scope, and shift the emphasis toward structural faithfulness to native high-order constraint templates and the inductive bias this provides.
> 4. Your suggestion on loss ablations is helpful. To test the appendix explanation of $\mathcal{L}\_{\mathrm{warm}}$, we added a loss ablation at the 17-19 givens sparsity limit. As shown in **Table 8** at anonymous URL https://anonymous.4open.science/r/rebuttal33745/, the full objective gives 91.80% puzzle accuracy, while removing $\mathcal{L}\_{\mathrm{warm}}$ drops it to 78.30%; removing the other two gives 87.10% and 88.20%, respectively. On cost and scalability, we added an inference-side complexity table for 9x9 and 16x16 Sudoku. **Table 7** at the same anonymous URL shows the optimization dimension increasing from 729 to 4096, together with increased inference latency and memory footprint. The optimization layer merits separate analysis and remains the key source of HONet's strict feasibility guarantees and effective geometric gradients under fixed symbolic structure. We will note differentiable first-order layers as a follow-up direction for their potential wall-clock benefit.
>
> We appreciate your suggestions, and hope these clarifications address your concerns about the paper's positioning, theoretical framing, and computational scalability more directly.

---

> > ### Author Rebuttal · Reviewer_JWnB · 2026-04-04
> >
> > Thank you for your response. While some of my comments are addressed I am still uncertain about:
> >
> > Prop 3.1 and the proof in the appendix. Would be nice to see it properly written up so we can carefully evaluate the technical content.
> >
> > Based on your response, I'm still uncertain whether the empirical results/contribution are sufficient here.

---

> > > ### Author Response · Authors · 2026-04-07
> > >
> > > Thank you very much for your follow-up. We agree that if Proposition 3.1 and its appendix proof are written out more explicitly, the technical scope we intend to convey will become clearer. To address this concern, while also respecting the space limit, we provide the main parts below.
> > >
> > > From a topological perspective, each constraint $c_k$ acts on a local variable subset $S_k\subseteq\mathcal V$ and corresponds to a local function $\Psi_k$ defined on $S_k$. For Exact Cover, define $\Psi_k(z_{S_k})=\lambda_{\min}$ if $\sum_{i\in S_k} z_i=1$, and $\Psi_k(z_{S_k})=\lambda_{\mathrm{penalty}}$ otherwise, where $\lambda_{\min}<\lambda_{\mathrm{penalty}}$.
> > >
> > > Let $G_{\mathrm{clq}}(\mathcal V,\mathcal E)$ denote the standard untyped clique expansion of the hypergraph representation $(\mathcal V,\mathcal E)$, i.e., the pairwise graph obtained by preserving only pairwise co-occurrence relations among variables while discarding native hyperedge identity/incidence information. Concretely, two variables are connected iff there exists some hyperedge $e\in\mathcal E$ such that they co-occur in $e$.
> > >
> > > **Proposition.** For any Exact Cover constraint $c_k$ with scope $S_k\subseteq\mathcal V$, the local function $\Psi_k$ is permutation-invariant on $S_k$. In particular, there exists a permutation-invariant hyperedge aggregation form that preserves the native scope and can represent $\Psi_k$. On the other hand, the standard untyped clique expansion mapping $(\mathcal V,\mathcal E)\mapsto G_{\mathrm{clq}}(\mathcal V,\mathcal E)$ is not injective with respect to the native higher-order constraint structure.
> > >
> > > **Proof.** Let $\mathcal V$ be the variable set and $\mathcal E$ the hyperedge set. For any $c_k\in\mathcal E$, let $S_k$ denote its scope.
> > >
> > > For the first part, by definition, $\Psi_k$ depends only on $\sum_{i\in S_k} z_i$, and is therefore independent of the ordering of variables within $S_k$. Hence, $\Psi_k$ is permutation-invariant on $S_k$.
> > >
> > > Now consider a permutation-invariant aggregation form that preserves the native scope:
> > > $$
> > > g_k(z_{S_k})=\phi_k\ \left(\sum_{i\in S_k}\psi(z_i)\right).
> > > $$
> > >
> > > Take $\psi(z_i)=z_i$, so that $\sum_{i\in S_k}\psi(z_i)=\sum_{i\in S_k} z_i$. Define $\phi_k(t)=\lambda_{\min}$ if $t=1$, and $\phi_k(t)=\lambda_{\mathrm{penalty}}$ otherwise. Then $g_k(z_{S_k})=\Psi_k(z_{S_k})$.
> > >
> > > Therefore, there exists a permutation-invariant hyperedge aggregation form that preserves the native scope and can represent this local Exact Cover function. In other words, $\Psi_k$ is representationally compatible with this class of hyperedge aggregation forms.
> > >
> > > For non-injectivity, let $\mathcal V=${1,2,3}, $\mathcal E_1=${{1,2,3}}, and $\mathcal E_2$={{1,2},{2,3},{1,3}}. Their standard untyped clique expansions are both $K_3$.
> > >
> > > Define $\mathcal F(\mathcal V,\mathcal E)=${$z\in${0,1}$^{|\mathcal V|}:\sum_{i\in S(c_k)} z_i=1,\ \forall c_k\in\mathcal E\$}, where $S(c_k)$ denotes the scope of constraint $c_k$.
> > >
> > > For $(\mathcal V,\mathcal E_1)$, the only constraint is $z_1+z_2+z_3=1$, and thus $\mathcal F(\mathcal V,\mathcal E_1)=${(1,0,0),(0,1,0),(0,0,1)}.
> > >
> > > For $(\mathcal V,\mathcal E_2)$, the constraints are $z_1+z_2=1$, $z_2+z_3=1$, and $z_1+z_3=1$. Summing the three equations gives
> > > $$
> > > 2(z_1+z_2+z_3)=3,
> > > $$
> > > which is impossible for binary variables. Hence $\mathcal F(\mathcal V,\mathcal E_2)=\varnothing$.
> > >
> > > Thus, $\mathcal F(\mathcal V,\mathcal E_1)\neq \mathcal F(\mathcal V,\mathcal E_2)$ while $G_{\mathrm{clq}}(\mathcal V,\mathcal E_1)=G_{\mathrm{clq}}(\mathcal V,\mathcal E_2)$, and therefore $(\mathcal V,\mathcal E)\mapsto G_{\mathrm{clq}}(\mathcal V,\mathcal E)$ is not injective. This proves the proposition.
> > >
> > > The proposition formalizes two representation-level properties: the local Exact Cover function on a native scope is permutation-invariant, and there exists a permutation-invariant hyperedge aggregation form that preserves this native scope and can represent it; standard untyped clique expansion is not injective, and thus can collapse distinct native higher-order constraint templates into the same pairwise graph. Accordingly, HONet’s structural advantage is better characterized as structural faithfulness to native higher-order constraint templates, together with the induced inductive bias.
> > >
> > > At the same time, we would also like to clarify that the paper is not trying to claim that it already covers a broader family of problems. Rather, by using exact-cover-style tasks as a relatively clean testbed, we aim to characterize how learning and optimization should divide their roles when the feasible region is already determined by known symbolic structure, and how this division affects data efficiency and the handling of relaxation-gap issues. We believe this provides a clearer and testable modeling perspective for learnable structured solving.
> > >
> > > Thank you again for your careful reading and feedback.

---

### Decision · Program_Chairs · 2026-04-30

**Decision:**

Accept (regular)

**Comment:**

This paper proposes HONet, a neuro-symbolic framework combining a hypergraph encoder with a differentiable QP layer, emphasizing learning objective parameters over a fixed feasible region. Reviewers agree the approach is technically sound, coherent, and demonstrates strong data efficiency.

The main concerns are limited empirical scope (mostly Sudoku-like tasks), scalability due to the optimization layer, and clarity of some theoretical components. The rebuttal improves presentation and clarifies the intended scope.

While the generality is somewhat limited, the paper offers a clear and novel perspective on learning under fixed symbolic constraints, with promising empirical results. I believe this focused but well-executed contribution is valuable to the community.